# LATTICE: LEARNING TO EFFICIENTLY COMPRESS THE MEMORY

## ABSTRACT

Attention mechanisms have revolutionized sequence learning but suffer from quadratic computational complexity. This paper introduces Lattice, a novel recurrent neural network (RNN) mechanism that leverages the inherent low-rank structure of K-V matrices to efficiently compress the cache into a fixed number of memory slots, achieving sub-quadratic complexity. We formulate this compression as an online optimization problem and derive a dynamic memory update rule based on a single gradient descent step. The resulting recurrence features a state- and input-dependent gating mechanism, offering an interpretable memory update process. The core innovation is the orthogonal update: each memory slot is updated exclusively with information orthogonal to its current state, hence incorporating only novel, non-redundant data, which minimizes the interference with previously stored information. The experimental results show that Lattice achieves the best perplexity compared to all baselines across diverse context lengths and model sizes.

## 1 INTRODUCTION

The attention mechanism (Vaswani et al., 2017b) has become a cornerstone of sequence modeling, offering significant advantages over traditional recurrent and convolutional approaches. By enabling models to dynamically attend to relevant parts of an input sequence while leveraging parallel computation, it effectively captures long-range dependencies and enables in-context learning. These strengths have driven its widespread adoption across various domains, including natural language processing (NLP) (Devlin et al., 2018; Radford et al., 2018), and computer vision (Dosovitskiy et al., 2020; Arnab et al., 2021). However, despite its effectiveness, the quadratic time and space complexity of attention limits its scalability in long sequence modeling. Additionally, its reliance on an unbounded cache leads to inefficient memory management, further limiting its applicability in resource-constrained settings. These challenges have motivated the development of alternative architectures that aim to retain the expressivity of Transformers while addressing its computational bottlenecks.

Sequence mixing approaches like state space models (SSMs) (Gu et al., 2021; 2020; 2022; 2020; Mehta et al., 2022) and linear attention variants (Katharopoulos et al., 2020; Choromanski et al., 2020) have recently gained renewed interest as promising alternatives to softmax attention. While traditional SSMs, with their inherent linear recurrent structure, offer parallelization during training, they often struggle to match the expressivity of standard attention. Linear attention methods reduce complexity by approximating the attention matrix but can sacrifice accuracy. More recently, input-dependent SSMs (Gu & Dao, 2023; Dao & Gu, 2024) and modern gated linear RNNs (Orvieto et al., 2023; De et al., 2024; Beck et al., 2024; Peng et al., 2025; Yang et al., 2024a) have demonstrated enhanced expressiveness and improved in-context learning while enabling parallelization through techniques like the associative scan (Blelloch, 1990; Smith et al., 2023; De et al., 2024). However, a fundamental challenge remains: their ability to efficiently compress and summarize information over very long contexts is often limited by their fixed-size hidden states (Arora et al., 2024). Moreover, their linear updates to memory lack efficient mechanisms for selective interaction between stored information and incoming keys, limiting their ability to discard irrelevant or redundant content dynamically. Non-linear recurrent networks have been revisited in recent works (Beck et al., 2024; Sun et al., 2024; Behrouz et al., 2024; 2025; Karami et al., 2025; Zhang et al., 2025), offering expressive sequence models. On the other hand, global convolutions (Romero et al., 2021; Li et al., 2022; Poli et al., 2023) and their input-dependent variants (Karami et al., 2019; Karami & Ghodsi, 2024) offer another

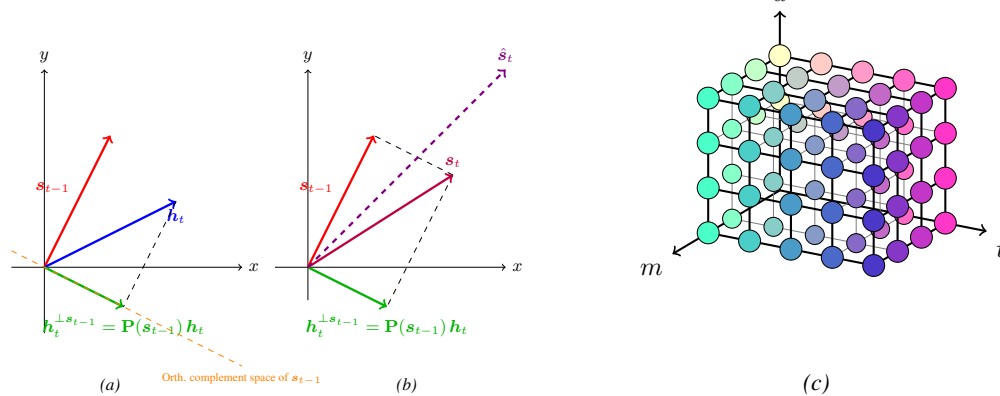

Figure 1: A geometric visualization of the proposed update rule. *(a)* A single current state vector, $\boldsymbol{s}_{t-1} = \mathbf{S}_{t-1}[:, i]$, an incoming token representation, $\boldsymbol{h}_t$, and its component orthogonal to the current state, $\boldsymbol{h}_t^{\perp \boldsymbol{s}_{t-1}}$. *(b)* Comparison of the updated state according to the proposed update rule ($\boldsymbol{s}_t = \boldsymbol{s}_{t-1} + \alpha_{i,t}\, \boldsymbol{h}_t^{\perp \boldsymbol{s}_{t-1}}$) and the updated state resulting from the superposition recurrence update of the standard linear attention ($\hat{\boldsymbol{s}}_t = \boldsymbol{s}_{t-1} + \alpha_{i,t}\, \boldsymbol{h}_t$, shown with a dashed arrow). For simplicity, a unit writing intensity ($\alpha_{i,t} = 1$) is assumed in both recurrent update rules. *(c)* Visualization of the relationships between $d \times m$ state matrices over time in state-dependent compression, depicted as interconnections of nodes in a 3D lattice. Each memory slot (state vector) is represented by a unique color.

direction for long context modeling by dynamically adapting convolutional filters to the input, but they are not inherently compatible with causal modeling, which is used in autoregressive language generation.

**Summary of contributions**   In this paper, we propose Lattice, a novel approach designed to address quadratic complexity of the attention layers. Our method compresses the cache into a fixed number of slots by leveraging the inherent low-rank structure of K-V matrices in an online optimization framework. This approach allows us to derive efficient recursive update rules for the memory (representing K-V associations) based on its existing state and the current token, resulting in sub-quadratic complexity. In contrast to existing SSMs/RNNs, which often rely on heuristics for memory management and lack explicit optimization for compression, we formulate the compression task as an optimization problem and use online gradient descent to drive the recurrent update rule for the memory, which results in an interpretable and expressive non-linear recurrent model. Lattice updates each memory slot exclusively with non-redundant information, specifically by incorporating only the component of the input token that is orthogonal to the current state of that memory slot.

## 2 BACKGROUND

For an input sequence $\mathcal{X} = [\boldsymbol{x}_1, \ldots, \boldsymbol{x}_T]$, where $\boldsymbol{x}_t \in \mathbb{R}^d$, the causal softmax attention mechanism generates output tokens $\boldsymbol{y}_t \in \mathbb{R}^d$, by attending to past tokens as:

$$\boldsymbol{y}_t = \mathcal{V}_t\, \texttt{Softmax}(\mathcal{K}_t^\top\, \boldsymbol{q}_t)\,. \tag{1}$$

Here, the queries, keys, and values are computed by linear projections of the input: $\boldsymbol{q}_t = \mathbf{W}_q\, \boldsymbol{x}_t$, $\boldsymbol{k}_t = \mathbf{W}_k\, \boldsymbol{x}_t$, $\boldsymbol{v}_t = \mathbf{W}_v\, \boldsymbol{x}_t$, where $\mathbf{W}_q, \mathbf{W}_k, \mathbf{W}_v \in \mathbb{R}^{d \times d}$ are learnable weight matrices. The key-value memory, represented by the caches $\mathcal{K}_t \in \mathbb{R}^{d \times t}$ and $\mathcal{V}_t \in \mathbb{R}^{d \times t}$, stacks the key and value vectors of each new token, leading to linearly growing caches. The retrieval of relevant information from this key-value cache can be rewritten as a weighted sum: $\boldsymbol{y}_t = \mathcal{V}_t\, \boldsymbol{a}_t$, where $\boldsymbol{a}_t = \texttt{Softmax}(\mathcal{K}_t^\top\, \boldsymbol{q}_t) \in \mathbb{R}^t$ Here, the vector $\boldsymbol{a}_t \in \mathbb{R}^t$ is the collection of the attention scores capturing correlations between $t$-th token and its historic context (past tokens). Hence, the attention in equation 1 can be seen as a non-linear query from an unbounded memory. The key-value cache size growth poses a significant memory bottleneck during inference, especially for long sequences. Additionally, each retrieval operation scales linearly with sequence length, resulting in an overall quadratic computational complexity $\mathcal{O}(T^2)$ for generating a full sequence of length $T$.

To address the computational and memory bottleneck of the Softmax attention, various alternatives have been proposed (Tay et al., 2022). A well-established approach involves employing the kernel trick to replace the softmax with a dot product of feature maps, $\phi(\boldsymbol{q}_t)$, $\phi(\boldsymbol{k}_t)$, (Katharopoulos et al., 2020), commonly known as *linear attention* (LA). The linear attention can be expressed as:

$y_t = \left( \sum_{i=1}^{t} \boldsymbol{v}_i \phi(\boldsymbol{k}_i)^\top \right) \phi(\boldsymbol{q}_t).$, which can be expressed as the following linear recurrent model, also known as input dependent state-space model (SSM) [1]:

$$\{\boldsymbol{y}_t\}_{t=1}^{T} = \mathrm{LA}(\{\boldsymbol{q}_t, \boldsymbol{k}_t, \boldsymbol{v}_t\}_{t=1}^{T}) := \begin{cases} \mathbf{S}_t = \mathbf{S}_{t-1} + \boldsymbol{v}_t\,\boldsymbol{k}_t^\top, & recurrence \\ \boldsymbol{y}_t = \mathbf{S}_t\boldsymbol{q}_t & memory\ read\text{-}out \end{cases} \quad (2)$$

This representation employs a simple linear recurrence to update the matrix-valued state $\mathbf{S}_t$, which compactly stores key-value associations memory at each time step. Importantly, the linearity is key to achieving sub-quadratic parallel computation during training, using methods such as chunkwise computation (Hua et al., 2022; Kacham et al., 2024) or parallel scan (Blelloch, 1990; Smith et al., 2023), while retaining a constant-time complexity per token at the inference.

Another approach to maintain bounded computational and memory requirements is to maintain a fixed-size key-value cache, where the memory matrices $\mathbf{K}, \mathbf{V} \in \mathbb{R}^{m \times d}$ are constrained to a fixed length $m \ll T$. A simple implementation of this idea is the sliding window attention which retains the most recent $m$ tokens by maintaining a first-in-first-out (FIFO) queue. While computationally efficient, sliding window attention suffers from a limited receptive field. This restricts the model's ability to capture long-range dependencies and maintain global context, resulting in a poor recall-memory trade-off (Arora et al., 2024). On the other hand, a growing body of research has observed that the key-value matrices in the attention often exhibit structured low-rank (Wang et al., 2020; Chen et al., 2021; Singhania et al., 2024). This insight suggests that instead of naively truncating memory, we can develop *efficient compression* techniques that selectively distill and store the essential context while discarding less relevant or redundant information.

The update rule in the (gated) linear attention, and its gated variant, typically relies on an additive outer product of input-dependent representations, which can be generally expressed as: $\mathbf{S}_t = \mathbf{S}_{t-1} + f_g(\boldsymbol{x}_t) \otimes f_v(\boldsymbol{x}_t)$ where $f_v(\boldsymbol{x}_t)$ is an embedding of the input token and $f_g$ can be interpreted as an input gate that controls the writing intensity.[2] While this *linear rank-one modification* to the state matrix (*a.k.a.* Hebbian-like update rule) enables efficient parallel computation, it suffers from a key limitation: the additive update term in the recurrence is not directly aware of the current memory state $\mathbf{S}_{t-1}$ and operates independently of it. This lack of state awareness can cause *key interference* and eventually lead to an *overcapacity regime* (Schlag et al., 2021), where multiple tokens attempt to write to the same memory slot when the size of memory is shorter than the sequence length.

Based on this insight, ideally, the writing intensity of the $t$-th token $\boldsymbol{x}_t$ to the $j$-th memory slot, $(\mathbf{S})_{j,:}$, should depend on the interaction between the new token itself and the content of that slot. From a gating perspective, the gating mechanism should have access to the current state of the memory to make informed decisions about which information to add or discard (Hochreiter et al., 1997; Gers et al., 2002). This requires a *state-dependent gating* mechanism that dynamically modulates updates based on the current memory state. Note that a naive conditioning on the state can break the ability to parallelize computations and hence it has been avoided in previous works. We will address this question through chunk-wise approximations later in the manuscript. In the following section, we frame the role of the recurrent layer as solving an online optimization problem and derive an optimal update rule to compress and retain essential information from a sequence.

## 3 COMPRESSION LAYER

**State-Dependent Compression for Unbounded Caches**   Our objective is to develop a compression model that dynamically updates and maintains a compact representation of the contextual history—encoded in the key and value caches of a transformer model—in a streaming manner. As new tokens arrive, the model selectively distills and stores essential contextual information into a compressed memory matrix. This enables computationally efficient querying, as the memory read-out is processed using the compressed state, i.e., $\boldsymbol{y}_t = \mathbf{S}_t\hat{\boldsymbol{r}}_t$ instead of querying from the full cache. Here, $\hat{\boldsymbol{r}}_t$ represents a retrieval vector analogous to the attention weights in the standard attention layer.

---

[1]As with many linear attention models, the normalization term, which can cause numerical instabilities (Qin et al., 2022), is dropped here. Furthermore, identity mapping is used as the feature map, effectively absorbing any transformation into the corresponding projection layers.

[2]These two are also called *role* and *filler* vectors in tensor product representation (Smolensky, 1990).

This lossy compression approach involves a trade-off between computational efficiency, memory usage, and query precision. A more compact memory representation (i.e., smaller $m$) reduces computational cost and memory footprint, but at the expense of information loss and lower fidelity in reconstructing the original context, thereby diminishing the overall expressivity of the model. We aim to design an optimal lossy compression layer that minimizes the precision loss. We can formulate this problem as reconstructing the input[3], where we enforce: $\mathbf{x}_t \approx \tilde{\mathbf{x}}_t = \mathbf{S}_t \boldsymbol{k}_t$, where $\tilde{\mathbf{x}}_t$ is the reconstructed input, $\mathbf{S}_t$ represents the dynamically updated memory matrix, and $\boldsymbol{k}_t$ is a latent representation vector[4].

Inspired by classical representation learning techniques such as dictionary learning, sparse coding, and structured matrix factorization (Mairal et al., 2009; Lyu et al., 2020)[5], we interpret our approach as dynamically learning and updating basis vectors (a.k.a. dictionary atoms) and their corresponding latent coefficients (analogous to sparse codes).

## 3.1 Decoding Layer

For each input sequence, we model a *decoding layer*, denoted as $g(\boldsymbol{k}_t; \mathbf{S}_t)$, which operates on the latent representation $\boldsymbol{k}_t$ and is parameterized by the state matrix $\mathbf{S}_t$. Unlike standard neural network layers, here we aim to dynamically update $\mathbf{S}_t$, over the course of a sequence, thereby effectively memorizing and encoding the historic context up to time $t$. This makes it a decoding layer with an *internal state*, or equivalently, a *fast decoding layer*. Specifically, each token embedding $\boldsymbol{v}_t$ is paired with its corresponding latent representation (code) $\boldsymbol{k}_t$, and the decoding function $g(\boldsymbol{k}_t; \mathbf{S}_t)$ aims to reconstruct $\boldsymbol{v}_t$. To achieve this, we formulate an optimization problem that minimizes a loss function, $\ell_2$—which quantifies the dissimilarity between the decoded output and the target vector—as its objective at each time step:

$$\mathcal{L}_t = \ell\big(g(\boldsymbol{k}_t; \mathbf{S}_t),\ \boldsymbol{v}_t\big),\ \ \mathbf{S}_t \in \mathbb{R}^{d \times m},\ \boldsymbol{v}_t \in \mathbb{R}^d,\ \boldsymbol{k}_t \in \mathbb{R}^m \tag{3}$$

This is referred to as *compression loss* throughout this paper. Here, the latent representation $\boldsymbol{k}_t$ is generated by a model-based encoder network, simply modeled as a linear projection of the input: $\boldsymbol{k}_t = \mathbf{W}_k \mathbf{x}_t$ where $\mathbf{W}_k \in \mathbb{R}^{m \times d_x}$ is a projection weight matrix. This weight remains fixed during the internal state updates and is trained jointly with the rest of the model parameters in the outer training loop. This setup aligns with meta-learning frameworks (Schmidhuber, 1992; Thrun & Pratt, 1998; Andrychowicz et al., 2016; Sun et al., 2024) or bilevel optimization approaches (Liu et al., 2022; Chen et al., 2022).

The proposed framework consists of two distinct types of parameters: (I) the internal states of the compression layers, $\mathbf{S}_t$, which dynamically store in-context information for each sequence, and (II) outer model parameters, including the projection layer weights, collectively denoted as $\mathcal{W}$, which capture the broader patterns in the training set. This leads to a bilevel learning process, composed of:

- *Inner Loop (State Update):* A fast update level that adapts the internal states $\mathbf{S}_t$ for each token within a sequence by minimizing the compression loss in equation 3. Each sequence effectively serves as a dataset for the inner loop, which encodes in-context information into a sequence of evolving states $\{\mathbf{S}_t\}_{t=1}^T$. Throughout this process, the outer model weights, $\mathcal{W}$, remain frozen.

- *Outer Training Loop:* The regular training of the neural network that learns $\mathcal{W}$ by minimizing the average loss across all training sequences for the (self-)supervised learning task. This slower loop typically employs standard optimizers such as ADAM (Kingma & Ba, 2014) and learns generalizable patterns in the training dataset.

From an optimization perspective, this bilevel process is analogous to alternating optimization (Goldstein et al., 2014), where the inner loop optimizes the state $\mathbf{S}_t$ while keeping $\mathcal{W}$ fixed, and the outer loop optimizes $\mathcal{W}$ based on the adapted states.

---

[3]Note that while the objective reconstructs from $\mathbf{S}_t$ only the current time step $\mathbf{x}_t$, due to the online gradient descent form of how this objective is minimized, the state $\mathbf{S}_t$ will compress and be able to reconstruct the entire sequence, i.e. $\mathbf{S}_t \boldsymbol{k}_k \approx \mathbf{x}_k$

[4]Since $\mathbf{S}_t$ has the same interpretation as the state in RNNs and SMMs, the same notation is reused.

[5]This problem has been studied under various names over the decades, including dictionary learning, factor analysis, topic modeling, and component analysis, each with slightly different constraints and emphases (Lyu et al., 2020).

The focus of this work is on designing an optimal update rule for the memory states. Due to its streaming nature, a standard approach for a sequence model is to treat the inner loop as an online regression problem and employ steepest descent. Specifically, the internal state is dynamically updated using a single gradient descent step per token:

$$\mathbf{S}_t = \mathbf{S}_{t-1} - \gamma_t \nabla_S \mathcal{L}(\mathbf{S}_{t-1}, \boldsymbol{v}_t, \boldsymbol{k}_t) \tag{4}$$

This recursive update yields a sequence of states $\{\mathbf{S}_t\}_{t=1}^T$, where each new state $\mathbf{S}_t$ is a nonlinear function of the current state and the current input tokens, ensuring a causal and context-dependent evolution of the internal state.

### 3.1.1 STATE NORMALIZATION

Following common practices in dictionary and subspace learning—where basis vectors (a.k.a. dictionary atoms or principal components) are normalized—we apply column-wise normalization to each state vector of the state matrix. Hence, the decoding function is defined using the normalized state matrix $\phi(\mathbf{S}_t)$ as:

$$\hat{\boldsymbol{v}}_t = g(\boldsymbol{k}_t; \mathbf{S}_t) = \phi(\mathbf{S}_t)\boldsymbol{k}_t.$$

At each time step, the internal states are updated to ensure the linear combination of normalized state vectors closely approximates the target vector $\boldsymbol{v}_t$. We explore two objectives to achieve this.

First, we adopt the standard $\ell_2$ reconstruction loss, which measures the squared Euclidean distance between the decoded vector and the target:

$$\mathcal{L}_t = \|\phi(\mathbf{S}_t)\boldsymbol{k}_t - \boldsymbol{v}_t\|^2, \quad \mathbf{S}_t \in \mathbb{R}^{d \times m}, \ \boldsymbol{v}_t \in \mathbb{R}^d, \ \boldsymbol{k}_t \in \mathbb{R}^m \tag{5}$$

To derive the closed-form gradient of this objective, let's define the normalized state matrix: $\Phi = [\phi_1, ..., \phi_m]$, where $\phi_i = \frac{\boldsymbol{s}_i}{\|\boldsymbol{s}_i\|}$ and $\boldsymbol{s}_i$ is the $i$-th column of $\mathbf{S}_{t-1}$ ($i$-th basis vector), and denote the reconstruction error as $\boldsymbol{e}_t := \phi(\mathbf{S}_{t-1})\boldsymbol{k}_t - \boldsymbol{v}_t$. The Jacobian of the normalization function is

$$\mathbf{J}_\phi(\boldsymbol{s}_i) = \frac{1}{\|\boldsymbol{s}_i\|}\left(\mathbf{I} - \frac{\boldsymbol{s}_i\boldsymbol{s}_i^\top}{\|\boldsymbol{s}_i\|^2}\right) = \frac{\mathbf{P}(\boldsymbol{s}_i)}{\|\boldsymbol{s}_i\|}.$$

Therefore, by applying the chain rule, the gradient of the loss with respect to $\mathbf{S}$ is given by:

$$\nabla_{\mathbf{S}}\mathcal{L}_t = \left[\boldsymbol{e}_t^\top \frac{\mathbf{P}(\boldsymbol{s}_1)}{\|\mathbf{s}_1\|}k_{t_1}, \quad \dots, \quad \boldsymbol{e}_t^\top \frac{\mathbf{P}(\boldsymbol{s}_m)}{\|\mathbf{s}_m\|}k_{t_m}\right] = (\boldsymbol{e}_t^\top \times_1 \mathcal{P}) \odot \boldsymbol{k}_t^\top \tag{6}$$

where $\odot$ denotes the element-wise (Hadamard) product with broadcasting, and $\times_1$ is a vector-tensor product defined as $\boldsymbol{e}^\top \times_1 \mathcal{P} := [\boldsymbol{e}^\top \mathcal{P}_{:,:,1}, \dots, \boldsymbol{e}^\top \mathcal{P}_{:,:,m}]$[6], while the tensor $\mathcal{P} := \left[\frac{\mathbf{P}(\boldsymbol{s}_1)}{\|\mathbf{s}_1\|}, \dots, \frac{\mathbf{P}(\boldsymbol{s}_m)}{\|\mathbf{s}_m\|}\right] \in \mathbb{R}^{d \times d \times m}$ is formed by stacking Jacobian matrices along the last dimension.

Alternatively, the compression objective can be formulated to maximize the dot-product similarity between the decoded output and target representation:

$$\mathcal{L}_t = -\langle\phi(\mathbf{S}_t)\boldsymbol{k}_t, \boldsymbol{v}_t\rangle, \quad \mathbf{S}_t \in \mathbb{R}^{d \times m}, \ \boldsymbol{v}_t \in \mathbb{R}^d, \ \boldsymbol{k}_t \in \mathbb{R}^m \tag{7}$$

The closed-form expression for the gradient of this loss can be derived similarly:

$$\nabla_{\mathbf{S}}\mathcal{L}_t = -\boldsymbol{v}_t^\top \times_1 \left[\frac{\mathbf{P}(\boldsymbol{s}_1)}{\|\mathbf{s}_1\|}, \dots, \frac{\mathbf{P}(\boldsymbol{s}_m)}{\|\mathbf{s}_m\|}\right] \odot \boldsymbol{k}_t^\top \tag{8}$$

This gradient derivation reveals a *highly interpretable and interesting update rule*. The matrix $\mathbf{P}(\boldsymbol{s}_i) = \mathbf{P}(\phi_i) := \left(\mathbf{I} - \frac{\boldsymbol{s}_i\boldsymbol{s}_i^\top}{\|\boldsymbol{s}_i\|^2}\right)$, which appears in (6) and (8), is known as the *projection matrix onto the orthogonal complement* of $\boldsymbol{s}_i$ in linear algebra (Strang, 2000, §3.3). This insight implies that the update for each memory slot (column $\boldsymbol{s}_i$) is driven by a projection of the input vector (e.g., $\boldsymbol{h}_t = -\boldsymbol{v}_t$ in (8) or $\boldsymbol{h}_t = \boldsymbol{e}_t$ in (6)) onto the space orthogonal to that slot. This suggests an interpretable decomposition of the $\boldsymbol{h}_t$ into two components: I) $\boldsymbol{h}_t^{\perp \boldsymbol{s}_i}$, The component orthogonal to $\boldsymbol{s}_i$, which is used to update the memory slot. II) $\boldsymbol{h}_t^{\| \boldsymbol{s}_i}$, The component of $\boldsymbol{h}_i$ aligned with $\boldsymbol{s}_i$, which is discarded in the update rule, ensuring non-redundant updates. This implies that *each memory slot updated only with new information that is not already captured in that slot*. The scalar $k_{i,t}$ acts as a *writing intensity*, determining the t-th token's contribution to the $i$-th memory slot. This orthogonal update process is visualized in Figure 1.

---

[6]This can be implemented using Einstein summation as `einsum("d1, d1 d m -> d m", e, J)`.

Therefore, applying online gradient descent (4) to the compression losses offers a principled approach for designing a recurrent update rule based on an orthogonal projection onto the current state. We unify these update rules into a general formulation, referred to as *Orthogonal State Recurrence (OSR)*:

$$\mathbf{S}_t = \mathbf{S}_{t-1} - \gamma_t \boldsymbol{h}_t^\top \times_1 \left[ \frac{\mathbf{P}(\boldsymbol{s}_1)}{\|\boldsymbol{s}_1\|}, \dots, \frac{\mathbf{P}(\boldsymbol{s}_m)}{\|\boldsymbol{s}_m\|} \right] \odot \boldsymbol{k}_t^\top \tag{9}$$

Here, $\boldsymbol{h}_t := \boldsymbol{e}_t$ and $\boldsymbol{h}_t := -\boldsymbol{v}_t$ for the $\ell_2$ loss and the dot-product similarity objective, respectively. Table 1 provides a summary comparing the online gradient descent-based recurrences corresponding to the proposed compression layers with those of existing RNNs. We also present an alternative encoding representation for the compression layer in Appendix B.1. In section 3.4 we provide a simplified form of the update.

## 3.2 STABILIZING MEMORY UPDATES VIA NORMALIZATION

At each recurrence step, a memory slot is updated by incorporating only the component of the new information that is *orthogonal* to its current state. Formally, we update the $i$-th memory slot as $\boldsymbol{s}_{i,t} = \boldsymbol{s}_{i,t-1} + \Delta \boldsymbol{s}_{i,t}$, where $\Delta \boldsymbol{s}_{i,t} := c_{i,t} \boldsymbol{h}_t^{\perp \boldsymbol{s}_{i,t-1}}$, with $\boldsymbol{h}_t^{\perp \boldsymbol{s}_{i,t-1}}$ denoting the component of the incoming token that is orthogonal to $\boldsymbol{s}_{i,t-1}$ and $c_{i,t}$ an input-dependent writing intensity. While this update scheme avoids interfering with the existing memory by *adding only novel, non-redundant information*, it leads to a monotonic increase in the norm of $\boldsymbol{s}_i$ with each update, as shown by the Pythagorean theorem: $\|\boldsymbol{s}_{i,t}\|^2 = \|\boldsymbol{s}_{i,t-1}\|^2 + \|\Delta \boldsymbol{s}_{i,t}\|^2$. This unbounded growth can cause numerical instability and state magnitude explosion or may dilute the effective representation of information over time.

To address this issue, we constrain the feasible set for the state vectors to the unit sphere $\mathcal{C} = \{\boldsymbol{s} \in \mathbb{R}^d \mid \|\boldsymbol{s}\| = 1\}$, and enforce this constraint by projecting the resulting Euclidean update back onto $\mathcal{C}$, denoted by $\mathcal{P}_{\mathcal{C}}(\cdot)$, at each time step. Therefore, the effective update becomes

$$\boldsymbol{s}_{i,t} = \mathcal{P}_{\mathcal{C}}(\boldsymbol{s}_{i,t-1} + \Delta \boldsymbol{s}_{i,t}) = \beta_{i,t} (\boldsymbol{s}_{i,t-1} + \Delta \boldsymbol{s}_{i,t}), \tag{10}$$

$$\text{where } \beta_{i,t} = \left(1 + \|\Delta \mathbf{s}_{i,t}\|^2\right)^{-\frac{1}{2}}, \text{ assuming } \|\mathbf{s}_{i,t-1}\| = 1.$$

This normalization step, achieved by multiplying with the scalar $\beta_{i,t}$, ensures that the updated state $\boldsymbol{s}_{i,t}$ remains on the unit sphere while preserving the steepest-descent direction, thus maintaining stability and allowing the model to effectively store relevant information. In addition, this normalization of the recurrence terms acts analogously to a forgetting gate in RNNs and also normalizes the step size of the update term—a technique known to improve convergence in optimization algorithms such as Adagrad (Duchi et al., 2011) and Adam (Kingma & Ba, 2014). In the following proposition, we formalize the relation between the proposed *Normalized Orthogonal State Recurrence* (NOSR) and Riemannian optimization (Absil et al., 2009; Boumal, 2023).

**Proposition 3.1 (Equivalence to Gradient Descent on Riemannian Manifold)** *Let $\mathcal{C} = \{\mathbf{s} \in \mathbb{R}^d \mid \|\mathbf{s}\| = 1\}$ be the unit sphere. Then, the projected gradient update of the form $\boldsymbol{s}_{i,t} = \mathcal{P}_{\mathcal{C}}(\boldsymbol{s}_{i,t-1} + \Delta \boldsymbol{s}_{i,t})$ (as in equation 10), where the update term $\Delta \boldsymbol{s}_{i,t}$ lies in the subspace orthogonal to $\boldsymbol{s}_{i,t-1}$ (cf. (9)), is equivalent to a retraction step in Riemannian optimization (Bonnabel, 2013).*

## 3.3 FORGETTING BY STATE REGULARIZATION

Similar to how regularization is used in standard neural network training to control the memorization of the model, we can apply regularization to the states in the inner loop to manage memory retention. Specifically, applying $\ell_2$ regularization to the state matrix $\mathbf{S}_t$ yields the regularized objective function: $\hat{\mathcal{L}}_t = \|\phi(\mathbf{S}_t) \boldsymbol{k}_t - \boldsymbol{v}_t\|^2 + \frac{\lambda_t}{2} \|\mathbf{S}_t\|_F^2$, where $\lambda_t$ is the regularization parameter and $\| \cdot \|_F$ denotes the Frobenius norm. Optimizing this differentiable objective using gradient descent results in a recurrence with state decay for the decoding compression layer (equation 5):

$$\mathbf{S}_t = \mu_t \mathbf{S}_{t-1} - \gamma_t \nabla_S \mathcal{L}(\mathbf{S}_{t-1}, \boldsymbol{v}_t, \boldsymbol{k}_t), \tag{11}$$

where the scalar $\mu_t = 1 - \gamma_t \lambda_t \in [0, 1]$ acts as a forget gate, controlling the proportion of the past memory that is retained in the update.

Table 1: Comparison of the objective functions and their corresponding online gradient descent updates for the proposed and existing RNNs. We include several linear RNNs for comparison: Linear-Attention(LA) (Katharopoulos et al., 2020), Mamba2 (Dao & Gu, 2024) and DeltaNet (Schlag et al., 2021; Yang et al., 2024c), GLA (Yang et al., 2024b), RWKV-6/7 (Peng et al., 2024; 2025), Gated-DeltaNet (Yang et al., 2024a), and TTT (Sun et al., 2024). It is worth noting that, the effective recurrent update of the compression layers after re-scaling becomes online gradient descent on Riemannian manifold: $\mathbf{S}_t = \mathbf{1}\boldsymbol{\beta}_t^\top \odot (\mathbf{S}_{t-1} + \Delta\mathbf{S}_t)$ (equation 10). LA can be interpreted as online gradient descent with a fixed step size ($\gamma_t = 1$); however, more flexible, input-dependent step sizes are frequently used in recent RNNs (Orvieto et al., 2023; Qin et al., 2024; Gu & Dao, 2023). Mamba2 and Gated-DeltaNet employ a forgetting gate, which is equivalent to performing online gradient descent with L2 regularization and regularization factor $\lambda_t$. In Mamba2, the forget gate is controlled by $\mu_t = 1 - \lambda_t$, and the reparameterization for the forget gate and step size of Gated-DeltaNet is discussed in (Wang et al., 2025). In RWKV-7, the regularizer is column-wise: $\frac{1}{2}\|\mathbf{S}_t^\top\|_{\text{diag}(\boldsymbol{\lambda}_t)}^2 := \frac{1}{2}\text{Tr}(\mathbf{S}_t^\top \text{diag}(\boldsymbol{\lambda}_t)\mathbf{S}_t)$, resulting in a diagonal-plus-low-rank transition matrix. Here, $\times_1$ denotes vector-tensor product defined as $\boldsymbol{e}^\top \times_1 [\mathbf{J}_1, \ldots, \mathbf{J}_m] = [\boldsymbol{e}^\top\mathbf{J}_1, \ldots, \boldsymbol{e}^\top\mathbf{J}_m]$.

| Method | Objective $\mathcal{L}_t$ | Online Gradient Descent Update |
|---|---|---|
| Linear-Attention | $-\langle\mathbf{S}_t\boldsymbol{k}_t, \boldsymbol{v}_t\rangle$ | $\mathbf{S}_t = \mathbf{S}_{t-1} + \boldsymbol{v}_t\boldsymbol{k}_t^\top$ |
| Mamba2 | $-\langle\mathbf{S}_t\boldsymbol{k}_t, \boldsymbol{v}_t\rangle + \frac{\lambda_t}{2}\|\mathbf{S}_t\|_2^2$ | $\mathbf{S}_t = \mu_t\mathbf{S}_{t-1} + \boldsymbol{v}_t\boldsymbol{k}_t^\top$ |
| RWKV-6 & GLA | $-\langle\mathbf{S}_t\boldsymbol{k}_t, \boldsymbol{v}_t\rangle + \frac{\boldsymbol{\lambda}_t}{2}\|\mathbf{S}_t\|_2^2$ | $\mathbf{S}_t = \text{diag}(\boldsymbol{\mu}_t)\mathbf{S}_{t-1} + \boldsymbol{v}_t\boldsymbol{k}_t^\top$ |
| DeltaNet | $\left\|\mathbf{S}_t\boldsymbol{k}_t - \boldsymbol{v}_t\right\|^2$ | $\mathbf{S}_t = \mathbf{S}_{t-1}(\mathbf{I} - \gamma_t\boldsymbol{k}_t\boldsymbol{k}_t^T) + \gamma_t\boldsymbol{v}_t\boldsymbol{k}_t^T$ |
| Gated-DeltaNet | $\left\|\mathbf{S}_t\boldsymbol{k}_t - \boldsymbol{v}_t\right\|^2 + \frac{\lambda_t}{2}\|\mathbf{S}_t\|_2^2$ | $\mathbf{S}_t = \mu_t\mathbf{S}_{t-1}(\mathbf{I} - \gamma_t\boldsymbol{k}_t\boldsymbol{k}_t^T) + \gamma_t\boldsymbol{v}_t\boldsymbol{k}_t^T$ |
| RWKV-7 | $\left\|\mathbf{S}_t\boldsymbol{k}_t - \boldsymbol{v}_t\right\|^2 + \frac{1}{2}\|\mathbf{S}_t^\top\|_{\text{diag}(\boldsymbol{\lambda}_t)}^2$ | $\mathbf{S}_t = \mathbf{S}_{t-1}(\text{diag}(\boldsymbol{\mu}_t) - \gamma_t\boldsymbol{k}_t\boldsymbol{k}_t^T) + \gamma_t\boldsymbol{v}_t\boldsymbol{k}_t^T$ |
| TTT | $\left\|\phi(\mathbf{S}_t\boldsymbol{k}_t) - \boldsymbol{v}_t\right\|^2$ | $\mathbf{S}_t = \mathbf{S}_{t-1} - \gamma_t\boldsymbol{e}_t^\top\frac{\mathbf{P}(\boldsymbol{z}_t)}{\|\boldsymbol{z}_t\|}\boldsymbol{k}_t^\top$ |
| Lattice (Dec) (5) | $\left\|\phi(\mathbf{S}_t)\boldsymbol{k}_t - \boldsymbol{v}_t\right\|^2$ | $\mathbf{S}_t = \mathbf{1}\boldsymbol{\beta}_t^\top \odot (\mathbf{S}_{t-1} - \gamma_t\boldsymbol{e}_t^\top \times_1 [\frac{\mathbf{P}(\boldsymbol{s}_1)}{\|\boldsymbol{s}_1\|}, \ldots, \frac{\mathbf{P}(\boldsymbol{s}_m)}{\|\boldsymbol{s}_m\|}] \odot \boldsymbol{k}_t^\top)$ |
| Lattice (Sim) (7) | $-\langle\phi(\mathbf{S}_t)^\top\boldsymbol{v}_t, \boldsymbol{k}_t\rangle$ | $\mathbf{S}_t = \mathbf{1}\boldsymbol{\beta}_t^\top \odot (\mathbf{S}_{t-1} + \gamma_t\boldsymbol{v}_t^\top \times_1 [\frac{\mathbf{P}(\boldsymbol{s}_1)}{\|\boldsymbol{s}_1\|}, \ldots, \frac{\mathbf{P}(\boldsymbol{s}_m)}{\|\boldsymbol{s}_m\|}] \odot \boldsymbol{k}_t^\top)$ |
| Lattice (Enc) (18) | $\left\|\phi(\mathbf{S}_t)^\top\boldsymbol{v}_t - \boldsymbol{k}_t\right\|^2$ | $\mathbf{S}_t = \mathbf{1}\boldsymbol{\beta}_t^\top \odot (\mathbf{S}_{t-1} - \gamma_t\boldsymbol{v}_t^\top \times_1 [\frac{\mathbf{P}(\boldsymbol{s}_1)}{\|\boldsymbol{s}_1\|}, \ldots, \frac{\mathbf{P}(\boldsymbol{s}_m)}{\|\boldsymbol{s}_m\|}] \odot \boldsymbol{e}_t^\top)$ |

**General Form.** By integrating all components, we arrive at the complete form for the non-linear state transition and memory read-out of Lattice:

$$\text{Lattice}(\{\boldsymbol{k}_t, \boldsymbol{v}_t, \boldsymbol{q}_t\}_{t=1}^T) = \begin{cases} \mathbf{S}_t = (\mu_t\mathbf{1}\boldsymbol{\beta}_t^\top)\odot\mathbf{S}_{t-1} - \gamma_t\boldsymbol{h}_t^\top\times_1\left[\frac{\mathbf{P}(\boldsymbol{s}_1)}{\|\boldsymbol{s}_1\|}, \ldots, \frac{\mathbf{P}(\boldsymbol{s}_m)}{\|\boldsymbol{s}_m\|}\right]\odot(\boldsymbol{k}_t\odot\boldsymbol{\beta}_t)^\top \\ \boldsymbol{y}_t = \mathbf{S}_t\boldsymbol{q}_t \end{cases}$$

where $\boldsymbol{\beta}_t \in \mathbb{R}^m$ is the per-slot normalization factor. We compare the proposed orthogonal state update with the delta rule linear recurrence (Widrow & Hoff, 1988) and nonlinear update rule of TTT (Sun et al., 2024) in Appendix B.3. In the following we present efficient computation of the Lattice and a chunk-wise parallelization.

### 3.4 Efficient Computation

The update rules presented in this work (equation 20) involves computing the projection $\boldsymbol{h}_t^{\perp\boldsymbol{s}_i} = \mathbf{P}\boldsymbol{h}_t$. Given the identity plus rank-one form of $\mathbf{P}$, the projection operation reduces to a dot product and a scalar-vector multiplication: $\boldsymbol{h}_t^{\perp\boldsymbol{s}_i} = \boldsymbol{h}_t - \frac{\boldsymbol{s}_i(\boldsymbol{s}_i^\top\boldsymbol{h}_t)}{\|\boldsymbol{s}_i\|^2}$, avoiding a full matrix-vector multiplication. The computational cost of each projection is linear in the state dimension $d$, leading to an overall complexity of $\mathcal{O}(d\,m)$ for the full recurrence in equation 9. By eliminating the need for vector-Jacobian-product (vjp) computations, this explicit form can be expressed as follows[7]:

$$\mathbf{S}_t = \mathbf{G}_t\odot\mathbf{S}_{t-1} + \underbrace{\mathbf{1}(\hat{\boldsymbol{h}}_t\odot\hat{\boldsymbol{k}}_t)^\top\odot\mathbf{S}_{t-1} - \boldsymbol{h}_t\hat{\boldsymbol{k}}_t^\top}_{-\gamma_t\nabla_S\mathcal{L}(\mathbf{S}_{t-1}, \boldsymbol{v}_t, \boldsymbol{k}_t)} \tag{12}$$

where $\mathbf{G}_t = \mu_t\mathbf{1}\boldsymbol{\beta}_t^\top \in \mathbb{R}^{d\times m}, \ \hat{\boldsymbol{h}}_t = \mathbf{S}_{t-1}^\top\boldsymbol{h}_t \in \mathbb{R}^m, \ \hat{\boldsymbol{k}}_t = \gamma_t\boldsymbol{\beta}_t\odot\boldsymbol{k}_t \in \mathbb{R}^m$

#### 3.4.1 Parallel and Hardware Efficient Form

Various methods have been explored to enable parallel evaluation of non-linear RNNs. One strategy, as proposed by Lim et al. (2023); Gonzalez et al. (2024), involves casting inference as finding the

---

[7]For notation brevity and due to the normalization step in equation 10, we assume $\|\boldsymbol{s}_{i,t-1}\| = 1$.

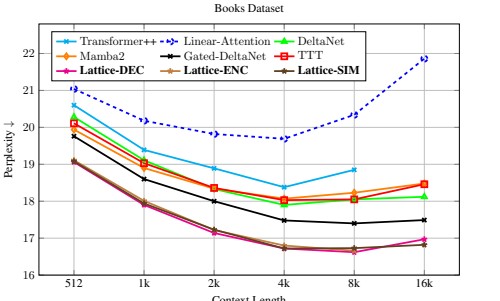 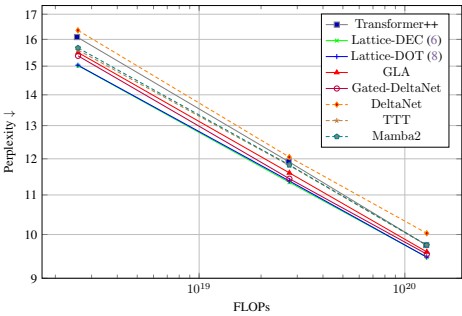

Figure 2: Scaling patterns. *(Left)*: Perplexity vs. context length (110M models, Books dataset). *(Right)*: Perplexity vs. model size (Chinchilla-style scaling, FineWeb-Edu). Transformer++ results are capped at $T \leq 8k$ since training them on very long contexts from scratch often performs poorly (Touvron et al., 2023).

solution to a fixed-point equation, thereby achieving parallelism. In a different approach, Sun et al. (2024) introduced a parallel chunk-wise solution using a gradient approximation. This method splits a sequence into non-overlapping chunks and utilizes the state at the beginning of each chunk to approximate the gradients for the entire chunk in parallel. We follow this approach and denote the state at the beginning of the chunk (i.e., the final state from the preceding chunk) by $\mathbf{S}_{t'}$, where $t' = t - \mathrm{mod}(t, C)$ with $C$ denoting the chunk size. The gradient is then approximated as $\nabla_S \mathcal{L}(\mathbf{S}_{t'}, \boldsymbol{v}_t, \boldsymbol{k}_t)$. This approximation linearizes the nonlinear recurrence (12) as:

$$\mathbf{S}_t = \mathbf{G}_t \odot \mathbf{S}_{t-1} + \mathbf{1}(\tilde{\boldsymbol{h}}_t \odot \hat{\boldsymbol{k}}_t)^\top \odot \mathbf{S}_{t'} - \boldsymbol{h}_t \hat{\boldsymbol{k}}_t^\top, \quad \text{where } \tilde{\boldsymbol{h}}_t = \mathbf{S}_{t'}^\top \boldsymbol{h}_t. \tag{13}$$

Now, for the time steps $t = bC + \tau$ in the $b$-th block, let $\mathbf{X}^b = \boldsymbol{x}[bC+1 : b(C+1)]$ denote the stacked input into the chunk-wise matrices and $\mathbf{X}_\tau^b = \boldsymbol{x}[bC + \tau]$ (similarly for other vectors such as $\boldsymbol{q}, \boldsymbol{h}, \boldsymbol{k}$), and define $\boldsymbol{a}_\tau^b = \prod_{i=bC+1}^{bC+\tau} \beta_i \mu_i$ and a block lower triangular tensor $\boldsymbol{\Omega}^b \in \mathbb{R}^{C \times C \times m}$ with components $\boldsymbol{\Omega}_{j,i,:}^b = \frac{\boldsymbol{a}_j^b}{\boldsymbol{a}_i^b} \mathbb{I}_{i \leq j}$ . Therefore, the layer update at the chunk-level is expressed as:

$$\mathbf{S}_b = \left(\mathbf{1}(\boldsymbol{a}_C^b + \boldsymbol{f}^b)^\top\right) \odot \mathbf{S}_{b-1} - \boldsymbol{H}^{b^\top}(\hat{\boldsymbol{K}}^b \odot \boldsymbol{\Omega}_{C,:,:}^b) \tag{14}$$
$$\boldsymbol{Y}^b = \left(\boldsymbol{Q}^b \odot (\boldsymbol{\Lambda}^b + \boldsymbol{F}^b)\right) \mathbf{S}_{b-1}^\top - \boldsymbol{P}^b \boldsymbol{H}^b$$

where $\boldsymbol{f}^b = \mathrm{diag}[\tilde{\boldsymbol{H}}^b(\hat{\boldsymbol{K}}^b \odot \boldsymbol{\Omega}_{C::}^b)]$ and $\boldsymbol{F}_{in}^b = \sum_{\tau=1}^C (\tilde{\boldsymbol{H}}^b \odot \hat{\boldsymbol{K}}^b)_{\tau n} \boldsymbol{\Omega}_{i\tau n}^b$ (refer to Appendix B for detailed `matmul` computation of $\boldsymbol{f}^b$, $\boldsymbol{F}^b$ and $\boldsymbol{\beta}_\tau^b$). Here, $\boldsymbol{P}^b \in \mathbb{R}^{C \times C}$ is a lower triangular matrix with $\boldsymbol{P}_{ij}^b = \sum_{k=1}^m \boldsymbol{Q}_{ik}^b \hat{\boldsymbol{K}}_{jk}^b \boldsymbol{\Omega}_{ijk}^b$. A `matmul`-optimal computation for $\boldsymbol{P}^b$ is presented in Zhang et al. (2024) using a sub-tiling technique. Furthermore, a closer inspection of (12) reveals that the recurrence can be simplified by linearizing only the $\hat{\boldsymbol{h}}_t$ term, leading to:

$$\mathbf{S}_t = \hat{\mathbf{G}}_t \odot \mathbf{S}_{t-1} - \boldsymbol{h}_t \hat{\boldsymbol{k}}_t^\top, \quad \text{where } \hat{\mathbf{G}}_t = \mathbf{1}(\mu_t \boldsymbol{\beta}_t + \tilde{\boldsymbol{h}}_t \odot \hat{\boldsymbol{k}}_t)^\top. \tag{15}$$

Here, $\hat{\mathbf{G}}_t$ is parameterized as a rank-one outer product, and hence this intra-chunk update can be computed efficiently using the parallel form of gated linear attention (GLA) (Zhang et al., 2024). More details on the parallel and hardware-efficient computation is presented in Appendix B.2.1.

## 4 EXPERIMENTS

**Setup:** We evaluate the proposed architecture across multiple language modeling tasks, benchmarking its performance on both short-context and long-context datasets. We trained models on different datasets: For language modeling and common-sense reasoning tasks, models are trained on FineWeb-Edu dataset (Penedo et al., 2024) with context length of 4k, for sequence length scaling pattern models are trained on Books3 dataset (a subset of The Pile dataset (Gao et al., 2020)) with different training context lengths ranging from 512 to 16k tokens (in increments of 2× per experiment). For all experiments, the training batch size is fixed at 0.5M tokens, irrespective of sequence length. We compare our method against Transformer++ model (Touvron et al., 2023) as well as the following sub-quadratic sequence models: Linear-Attention (LA) (Katharopoulos et al., 2020), TTT (Sun et al., 2024), DeltaNet (Yang et al., 2024c), Gated DeltaNet (Yang et al., 2024a), Mamba2 Dao & Gu (2024) and GLA (Zhang et al., 2024). Experimental details, including a training throughput comparison (Figure 8), are provided in Appendix C.

Table 2: Performance comparison on LM and zero-shot common-sense reasoning tasks. Models are trained on FineWeb-Edu dataset.

| Model | PIQA | Hella. | Wino. | ARC-e | ARC-c | CSQA | BoolQ | Avg. |
|---|---|---|---|---|---|---|---|---|
| | acc ↑ | acc_n ↑ | acc ↑ | acc ↑ | acc_n ↑ | acc ↑ | acc ↑ | |
| *340M params / 15B tokens* | | | | | | | | |
| Transformer++ | 66.76 | 40.40 | 52.38 | 49.47 | 27.04 | 33.01 | 58.93 | 46.93 |
| GLA | 67.52 | 42.10 | 52.09 | 53.11 | 29.36 | 36.77 | 59.79 | 48.68 |
| Mamba2 | 67.08 | 41.30 | 52.25 | 51.63 | 29.19 | 36.28 | 60.98 | 48.39 |
| DeltaNet | 66.21 | 40.80 | 52.64 | 50.74 | 27.73 | 36.20 | 62.05 | 48.05 |
| TTT | 66.59 | 41.10 | 51.86 | 52.14 | 26.67 | 35.71 | 60.89 | 47.88 |
| Gated-DeltaNet | 68.12 | 42.93 | 52.33 | 53.19 | 29.44 | 36.45 | 57.74 | 48.60 |
| Lattice-DEC (6) C=1 | 68.28 | 43.33 | 51.70 | 53.66 | 28.41 | 37.27 | 60.70 | 49.05 |
| Lattice-DEC (6) C=4 | 67.25 | 43.20 | 51.93 | 54.33 | 29.70 | 36.12 | 61.44 | 49.14 |
| Lattice-DOT (8) C=4 | 67.95 | 43.13 | 51.07 | 53.91 | 28.50 | 38.33 | 57.71 | 48.66 |
| *760M params / 30B tokens* | | | | | | | | |
| Transformer++ | 69.59 | 51.13 | 53.83 | 56.96 | 30.30 | 39.64 | 62.11 | 51.94 |
| GLA | 69.64 | 52.17 | 53.20 | 59.07 | 32.62 | 41.20 | 60.12 | 52.57 |
| Mamba2 | 70.18 | 50.67 | 52.64 | 59.28 | 33.99 | 41.77 | 57.19 | 52.25 |
| DeltaNet | 69.86 | 48.60 | 51.85 | 58.10 | 32.36 | 40.05 | 58.56 | 51.34 |
| TTT | 70.08 | 50.67 | 52.09 | 58.65 | 33.73 | 41.28 | 60.43 | 52.42 |
| Gated-DeltaNet | 71.38 | 51.87 | 53.99 | 61.06 | 34.08 | 39.97 | 56.18 | 52.64 |
| Lattice-DEC (6) | 71.00 | 52.93 | 53.28 | 59.66 | 34.68 | 43.90 | 59.33 | 53.54 |
| Lattice-DOT (8) | 71.76 | 51.70 | 54.38 | 59.49 | 36.57 | 43.57 | 60.73 | 54.03 |

Table 3: Ablation study of Lattice's components (110M parameters, trained on FineWeb-Edu). Each row starting with + adds a new component to the configuration in the row above it, beginning from the DeltaNet baseline. The final row represents the full Lattice configuration. By default $m = d = 64$ in all models.

| Configuration | ppl ↓ |
|---|---|
| DeltaNet | 16.35 |
| TTT C=1 | 15.60 |
| Lattice C=1 | |
| + orthogonal recurrence (6) | 15.38 |
| + normalized projection (10) | 15.07 |
| + forget-gate (11) | 15.02 |
| Lattice C=4 approx. (13) | 15.15 |
| Lattice C=4 approx. (15) | 15.12 |
| Lattice m=16 | 15.52 |
| Lattice m=32 | 15.26 |
| Lattice m=128 | 14.84 |
| Lattice m=192 | 14.71 |

The results in Figure 2 (and Figure 5) demonstrate that Lattice consistently achieves the best perplexity compared to all baselines across a range of context lengths. Importantly, the performance gains of Lattice relative to other linear RNNs become more pronounced as the sequence length grows. This trend highlights the effectiveness of the proposed approach for long-context modeling. Moreover, we show the scaling pattern of Lattice in Figure 2 with respect to increasing model size, showing that Lattice achieves lower perplexity than the baselines at different scales.

**Common-sense reasoning.** In Table 2, we report the zero-shot accuracy of the trained models on various commonsense reasoning tasks—including PIQA (Bisk et al., 2020), HellaSwag (Hella.) (Zellers et al., 2019), WinoGrande (Wino.) (Sakaguchi et al., 2021), ARC-easy (ARC-e) and ARC-challenge (Arc-c) (Clark et al., 2018), and Commonsense QA (CSQA) (Talmor et al., 2018), commonly used for LM benchmarking (Zhang et al., 2024; Yang et al., 2024c). As the results demonstrate, Lattice outperforms the baseline models on most of these tasks, achieving the highest average accuracy.

**Ablation.** In this study, we ablate key components of the Lattice to evaluate the contribution of each to the overall performance. The results in Table 3 underscore the significance of state normalization (8) and normalized projection (10) to the model's overall performance. Furthermore, our analysis indicates that the forget gate's impact on the overall performance is negligible, suggesting that the normalized projection introduced in (10) inherently acts as a forgetting mechanism. Finally, Lattice with only a quarter of memory slots ($m = 16$) outperforms TTT with $m = d = 64$, validating the proposed orthogonal update mechanism utilizes the memory capacity significantly more efficiently.

# 5 CONCLUSION

This work introduced a novel recurrent neural network mechanism designed for efficient information compression into a matrix-valued state with a limited number of memory slots. We approached this problem by framing it as an online optimization problem, deriving the memory's dynamic update rule from a single gradient descent step. The resulting recurrence features a state- and input-dependent gating mechanism, leading to an interpretable memory update process. A core feature of this mechanism is that each memory slot is updated exclusively with information that is orthogonal to its current state. This orthogonal update ensures that only new, non-redundant data is written into memory and minimizes the interference with previously stored information. Furthermore, the update includes an input- and state-dependent writing intensity, providing fine-grained control over the magnitude of the information written to each memory slot. With its sub-quadratic complexity, this mechanism offers a promising alternative to Transformers for pre-training or a method for efficiently fine-tuning pre-trained Transformers into RNNs.

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

## BROADER IMPACTS

This paper presents work whose goal is to advance the field of Machine Learning. There are many potential societal consequences of our work, none which we feel must be specifically highlighted here.

## A  DISCUSSION AND RELATED WORKS

**Fast Weight Programmers and Test-Time Training.**  The two-stage learning process adopted in our work draws inspiration from the concept of Fast Weight Programmers (FWPs) (Schmidhuber, 1992; Schlag et al., 2021) where a "slow" network dynamically updates the parameters of a "fast" network. In our framework, the compression layer in the inner loop can be seen as the fast network, with its memory states, $\mathbf{S}_t$, acting as "fast weights" that are rapidly adapted to the evolving contextual information. The outer loop, conversely, learns the generalizable parameters of the slow neural network, optimized across the entire training dataset. The continual reprogramming of fast network weights by slow models (Irie et al., 2021; Clark et al., 2022) is broadly recognized as Fast Weight Programming, also referred to as synaptic modulation (Von Der Malsburg, 1994) or input-dependent parameterization (Karami et al., 2019; Gu & Dao, 2023; Karami & Ghodsi, 2024), a technique known to enhance model expressiveness. In our architecture, the parameterization of the linear projections by the slow network facilitates this fast adaptation within the inner loop. Similarly, *Test-Time Training* (Sun et al., 2020; 2024; Behrouz et al., 2024; von Oswald et al., 2025) is a paradigm where a model adapts to each test instance by optimizing a self-supervised objective before making predictions. Our compression layer effectively implements a form of test-time training by dynamically updating its state based on the contextual information of the input sequence during inference. In contrast to the aforementioned works, our approach introduces an explicit learning mechanism for the "fast" compression layer, leading to an interpretable update rule for its internal states that optimally compresses the latest token into memory at test time.

Test-time training for sequence models have recently been formalized through the lens of online optimization. The works such as  Sun et al. (2024); Liu et al. (2024); Behrouz et al. (2024; 2025); Wang et al. (2025); Karami et al. (2025); von Oswald et al. (2025); Zhang et al. (2025) fall under this category. They have demonstrated that deriving recurrent update rules from the online optimization of a regression objective can yield powerful sequence models.

**Adaptive Filters.**  Classical adaptive filtering algorithms (Haykin, 2002) iteratively update their weights to minimize prediction error while efficiently adapting to streaming, non-stationary data. These methods share core principles with the online learning and dynamic memory updates employed in our work. In particular, the gradient descent-based update rules we adopted for memory adaptation are closely related to the Least Mean Squares (LMS) algorithm—also known as the Widrow-Hoff algorithm (Widrow & Hoff, 1988)—which updates weights using the instantaneous gradient of the squared error. Furthermore, variations such as Normalized Least Mean Squares (NLMS), which involves a normalized step size for improved convergence, and Leaky LMS, which incorporates a leakage factor used to prevent unbounded growth of filter weights, find parallels in our use of normalization mechanisms to stabilize memory update (Equation 10) and state decay (Equation 11). While these adaptive filtering methods rely on linear weight updates, our approach introduces a non-linear memory update rule that incorporates only the non-redundant components of the new token.

**Matrix Factorization**  Matrix factorization and dictionary learning are classical representation learning techniques that aim to extract essential features from complex data by approximating it as a linear combination of a reduced set of basis vectors, also known as dictionary atoms. This concept is also conceptually related to topic modeling, where the objective is to extract important features (topics) from a complex dataset to obtain a reduced representation (Blei, 2009; 2012). Mairal et al. (2010) proposed an online optimization algorithm for structured matrix factorization and sparse coding for i.i.d. stream of data, which efficiently scales to large datasets. Subsequently, Lyu et al. (2020) extended this work by proving the convergence of such an online algorithm in non-i.i.d. settings, where the sequential data forms a Markov chain. In a related area, Karami et al. (2017) formulated the identification of SSMs (a.k.a. linear dynamical systems) as a multi-view matrix factorization problem and proposed a convex optimizer for its solution. In contrast to the online matrix factorization in Mairal et al. (2009), which employs a model-free method to learn the latent coefficients (codes) and leverages block coordinate descent for optimization, our method formulates the memory update as a fast internal optimization procedure. We incorporate a simple encoding layer to generate the latent representation, $\boldsymbol{k}_t$, and integrate it into a larger deep neural network training procedure.

**Recent Advancements in RNNs.** More broadly, the field is witnessing a resurgence of interest in RNN research, with many new architectures emerging as viable alternatives to Transformers. Works such as Sun et al. (2023); Lin et al. (2025); Merrill et al. (2024); Kacham et al. (2024); Peng et al. (2023; 2025); Guo et al. (2025); Du et al. (2025); Siems et al. (2025); Hu et al. (2025) tackle the limitations of RNNs, introducing new mechanisms to improve long-range dependency, handling scalability, or efficient parallelizability.

A recent concurrent work, Large Chunk Test-Time Training (LaCT) (Zhang et al., 2025), focuses on maximizing hardware utilization for TTT layers. LaCT proposes using very large chunk sizes to maximize GPU utilization, which enables the use of computationally intensive optimizers like Muon—a advanced optimizer that performs spectral normalization (via Newton-Schulz orthogonalization) on the gradient matrix to speed up convergence. In contrast to LaCT's focus on hardware-centric optimization and inter-chunk normalization, our work focuses on the memory update mechanism itself. We derive the Orthogonal State Recurrence (OSR) from a state-normalized reconstruction objective, which ensures non-redundant information storage via input-dependent orthogonal projection. Furthermore, while LaCT applies normalization between chunks, Lattice enforces normalization at every recurrence step to structurally prevent state explosion, caused by the Pythagorean addition inherent in orthogonal updates, and induce an implicit gating mechanism.

**Remark A.1 (Delta Rule)** *Removing the non-linearity $\phi(\cdot)$ from the compression layer simplifies the online gradient descent update rule in (4) to*

$$\mathbf{S}_t = \mathbf{S}_{t-1} - \gamma_t(\mathbf{S}_{t-1}\boldsymbol{k}_t - \boldsymbol{v}_t)\boldsymbol{k}_t^\top = \mathbf{S}_{t-1}(\mathbf{I} - \gamma_t\boldsymbol{k}_t\boldsymbol{k}_t^\top) + \gamma_t\boldsymbol{v}_t\boldsymbol{k}_t^\top \tag{16}$$

*This linear update rule recovers the delta rule (Widrow & Hoff, 1988), known for its higher memory capacity (Prados & Kak, 1989) and has been demonstrated as an effective form of linear recurrence, particularly in associative recall tasks (Schlag et al., 2021; Yang et al., 2024c). Similar to linear transformers, the second term writes into memory via the outer product $\boldsymbol{v}_t\boldsymbol{k}_t^\top$, while the first term implements a forgetting mechanism, controlled by the new key $\boldsymbol{k}_t$, to remove old information from memory. Here, we propose a more efficient update rule based on the nonlinear interactions between the memory and the non-redundant information of the new keys.*

**Remark A.2 (Layer Normalization)** *In the proposed compression layers, normalization was applied to each state column. Alternatively, Sun et al. (2024) proposed applying normalization on the output of the decoding layer.[8] In this case, the decoding function becomes: $\hat{\boldsymbol{v}}_t = g(\boldsymbol{k}_t; \mathbf{S}_t) = \phi(\mathbf{S}_t\boldsymbol{k}_t)$. This formulation is analogous to applying* layer normalization *as commonly used in deep neural networks. As before, we can simplify the gradient $\nabla_{\mathbf{S}}\mathcal{L}_t$ to derive an interpretable update rule. Specifically, let $\phi(\boldsymbol{z}_t) = \frac{\boldsymbol{z}_t}{\|\boldsymbol{z}_t\|}$, where $\boldsymbol{z}_t := \mathbf{S}_{t-1}\boldsymbol{k}_t$, and define the reconstruction error as $\boldsymbol{e}_t := \hat{\boldsymbol{v}}_t - \boldsymbol{v}_t$. Applying the chain rule, we obtain:*

$$\frac{\partial\mathcal{L}_t}{\partial\mathbf{S}} = \boldsymbol{e}_t^\top \mathbf{J}_\phi(\boldsymbol{z}_t)\boldsymbol{k}_t^\top, \text{ where } \mathbf{J}_\phi(\boldsymbol{z}_t) = \frac{\mathbf{P}(\boldsymbol{z}_t)}{\|\boldsymbol{z}_t\|} = \frac{1}{\|\boldsymbol{z}_t\|}\left(\mathbf{I} - \frac{\boldsymbol{z}_t\boldsymbol{z}_t^\top}{\|\boldsymbol{z}_t\|^2}\right)$$

*Subsequently, the gradient descent update follows a nonlinear recurrence:*

$$\mathbf{S}_t = \mathbf{S}_{t-1} - \gamma_t\boldsymbol{e}_t^\top \frac{\mathbf{P}(\boldsymbol{z}_t)}{\|\boldsymbol{z}_t\|}\boldsymbol{k}_t^\top \tag{17}$$

*This nonlinear state recurrence incorporates an outer-product correction based on the projection of the reconstruction error $\boldsymbol{e}_t$ onto the orthogonal complement space of $\hat{\boldsymbol{v}}_t$.*

*The concept of normalizing the state vectors in our compression model, as described in §3.1, shares similarities with weight normalization techniques used in deep learning literature (Salimans & Kingma, 2016). Furthermore, the two interpretations presented above—applying normalization to the output of the decoding layer versus normalizing the state vectors—offer insights into the rationale behind different normalization schemes commonly used in deep learning, such as weight normalization and layer normalization (Ba, 2016). Each normalization method plays a distinct role in stabilizing training and improving generalization.*

---

[8]While the general formulation of TTT (Sun et al., 2024) applies a non-linearity to $\boldsymbol{z}_t$, their implementation specifically utilizes normalization.

## B DETAILED DERIVATIONS AND MODEL VARIANTS

### B.1 ENCODING LAYER

Principal Component Analysis (PCA) can be formulated as a linear regression problem, where the data is projected onto a lower-dimensional latent space (Goodfellow et al., 2016, §5.8) Inspired by this regression perspective, we define a encoding layer as: $\hat{\boldsymbol{k}}_t = f(\boldsymbol{v}_t; \mathbf{S}_t) = \phi(\mathbf{S}_t)^\top \boldsymbol{v}_t$ with the corresponding $\ell_2$ loss:

$$\mathcal{L}_t = \left\| \phi(\mathbf{S}_t)^\top \boldsymbol{v}_t - \boldsymbol{k}_t \right\|^2, \quad \mathbf{S}_t \in \mathbb{R}^{d \times m}, \ \boldsymbol{v}_t \in \mathbb{R}^d, \ \boldsymbol{k}_t \in \mathbb{R}^m \tag{18}$$

From this, we can derive a closed-form expression for the gradient, resulting in the recurrence:

$$\mathbf{S}_t = \mathbf{S}_{t-1} - \gamma_t \boldsymbol{v}_t^\top \times_1 \left[ \frac{\mathbf{P}(\boldsymbol{s}_1)}{\|\mathbf{s}_1\|}, \ldots, \frac{\mathbf{P}(\boldsymbol{s}_m)}{\|\mathbf{s}_m\|} \right] \odot \boldsymbol{e}_t^\top \tag{19}$$

**General form.** The formulations presented in Eqs. (6, 19, and 8) offer principled approaches for designing compression layers in our framework. In general, we refer to this update rule as *Orthogonal State Recurrence (OSR)*, which unifies compression layer into a common framework formulated as follows

$$\{\boldsymbol{y}_t\}_{t=1}^T = \mathrm{Lattice}(\{\boldsymbol{k}_t, \boldsymbol{v}_t, \boldsymbol{q}_t\}_{t=1}^T) := \begin{cases} \mathbf{S}_t = \mathbf{S}_{t-1} - \gamma_t \boldsymbol{h}_t^\top \times_1 \left[ \frac{\mathbf{P}(\boldsymbol{s}_1)}{\|\boldsymbol{s}_1\|}, \ldots, \frac{\mathbf{P}(\boldsymbol{s}_m)}{\|\boldsymbol{s}_m\|} \right] \odot \boldsymbol{c}_t^\top \\ \boldsymbol{y}_t = \mathbf{S}_t \boldsymbol{q}_t \end{cases}$$
$$\tag{20}$$

Here, the definitions of $\boldsymbol{h}_t$ and $\boldsymbol{c}_t$ vary depending on the specific layer:

$$\begin{cases} \{\boldsymbol{h}_t = \boldsymbol{e}_t, & \boldsymbol{c}_t = \boldsymbol{k}_t\}, & \text{Decoding Layer} \\ \{\boldsymbol{h}_t = \boldsymbol{v}_t, & \boldsymbol{c}_t = \boldsymbol{e}_t\}, & \text{Encoding Layer} \\ \{\boldsymbol{h}_t = -\boldsymbol{v}_t, & \boldsymbol{c}_t = \boldsymbol{k}_t\}, & \text{Similarity Objective} \end{cases}$$

Table 1 provides a summary comparing the online gradient descent-based recurrent corresponding to the proposed compression layers and those of existing RNNs.

### B.2 FORGETTING BY STATE REGULARIZATION

Alternative to $\ell_2$ regularization, we can induce sparsity in the memory states by applying element-wise $\ell_1$ norm [9], resulting in the following objective function:

$$\hat{\mathcal{L}}_t = \|\phi(\mathbf{S}_t) \boldsymbol{k}_t - \boldsymbol{v}_t\|^2 + \lambda_t \|\mathbf{S}_t\|_1. \tag{21}$$

This non-differentiable composite objective can be efficiently optimized using the *Proximal Gradient Descent Algorithm*, which iteratively performs a gradient descent with the smooth component and then applies the proximal operator associated with the non-differentiable regularizer (Parikh et al., 2014). This iterative procedure, commonly known as the Iterative Shrinkage-Thresholding Algorithm (ISTA) (Parikh et al., 2014; Beck & Teboulle, 2009), yields the update rule :

$$\mathbf{S}_t = \mathrm{prox}_{\gamma_t \lambda_t \|\cdot\|_1} \left( \mathbf{S}_{t-1} - \gamma_t \nabla_S \|\phi(\mathbf{S}_t) \boldsymbol{k}_t - \boldsymbol{v}_t\|^2 \right), \tag{22}$$

where the proximal operator for the $\ell_1$ norm corresponds to the *shrinkage (soft thresholding) operation*, defined as: $\mathrm{prox}_{\mu_t \|\cdot\|_1}(x) = \mathrm{sign}(x) \max(|x| - \mu_t, 0)$. By suppressing small values, this recurrence promotes sparsity in the learned memory representations.

#### B.2.1 PARALLEL AND HARDWARE EFFICIENT FORM

Various methods have been explored to enable parallel evaluation of non-linear RNNs. One strategy, as proposed by Lim et al. (2023); Gonzalez et al. (2024), involves casting inference as finding the solution to a fixed-point equation, thereby achieving parallelism. In a different approach, Sun et al. (2024) introduced a parallel chunk-wise solution using a gradient approximation. This method splits a sequence into non-overlapping chunks and utilizes the state at the beginning of each chunk to

---

[9]The $\ell_1$ norm is a relaxed version of the hard sparsity constrain, which drives small states toward zero.

approximate the gradients for the entire chunk in parallel. Following this technique, we let $\mathbf{S}_{t'}$ represent the state at the beginning of the chunk (i.e., the final state from the preceding chunk), where $t' = t - \text{mod}(t, C)$ with $C$ denoting the chunk size. The gradient is then approximated as $\nabla_S \mathcal{L}(\mathbf{S}_{t'}, \boldsymbol{v}_t, \boldsymbol{k}_t)$. This approximation linearizes the nonlinear recurrence (12) as:

$$\mathbf{S}_t = \mathbf{G}_t \odot \mathbf{S}_{t-1} + \mathbf{1}(\tilde{\boldsymbol{h}}_t \odot \hat{\boldsymbol{k}}_t)^\top \odot \mathbf{S}_{t'} - \boldsymbol{h}_t \hat{\boldsymbol{k}}_t^\top, \tag{23}$$

where $\tilde{\boldsymbol{h}}_t = \mathbf{S}_{t'}^\top \boldsymbol{h}_t$. Now, for the time steps $t = bC + \tau$ in the $b$-th block, let $\mathbf{X}^b = \boldsymbol{x}[bC + 1 : b(C + 1)]$ denote the stacked input into the chunk-wise matrices and $\mathbf{X}_\tau^b = \boldsymbol{x}[bC + \tau]$ (similarly for other vectors such as $\boldsymbol{q}, \boldsymbol{h}, \boldsymbol{k}$), and define the local cumulative product of decay factors as $\boldsymbol{a}_\tau^b = \prod_{i=bC+1}^{bC+\tau} \beta_i \mu_i$. We also define a block lower triangular tensor $\boldsymbol{\Omega}^b \in \mathbb{R}^{C \times C \times m}$ with components $\boldsymbol{\Omega}_{j,i,:}^b = \frac{\boldsymbol{a}_j^b}{\boldsymbol{a}_i^b} \mathbb{I}_{i \leq j}$ that are segmented cumulative product over the sub-block steps $i$ to $j$ ($1 \leq i \leq j \leq C$). Here $\mathbb{I}_{i \leq j}$ is the indicator function (equal to 1 if $i \leq j$ and 0 otherwise), the division $\frac{\boldsymbol{a}_j^b}{\boldsymbol{a}_i^b}$ is performed element-wise. Therefore, the recurrence equation 13 can be expressed at the chunk-level as:

$$\mathbf{S}_b = \left(\mathbf{1}(\boldsymbol{a}_C^b + \boldsymbol{f}^b)^\top\right) \odot \mathbf{S}_{b-1} - \boldsymbol{H}^{b^\top}(\hat{\boldsymbol{K}}^b \odot \boldsymbol{\Omega}_{C,:,:}^b) \tag{24}$$

$$\boldsymbol{Y}^b = \left(\boldsymbol{Q}^b \odot (\boldsymbol{\Lambda}^b + \boldsymbol{F}^b)\right) \mathbf{S}_{b-1}^\top - \boldsymbol{P}^b \boldsymbol{H}^b$$

where $\boldsymbol{f}^b = \text{diag}[\tilde{\boldsymbol{H}}^b(\hat{\boldsymbol{K}}^b \odot \boldsymbol{\Omega}_{C::}^b)] \in \mathbb{R}^m$, and $\boldsymbol{F}^b \in \mathbb{R}^{C \times m}$ is a matrix with entries $\boldsymbol{F}_{in}^b = \sum_{\tau=1}^{C}(\tilde{\boldsymbol{H}}^b \odot \hat{\boldsymbol{K}}^b)_{\tau n} \boldsymbol{\Omega}_{i\tau n}^b \; \forall 0 < i \leq C, \; 0 < j \leq m$. Both of these can be computed using *tensor contraction* or Einstein summation operation as

$$\boldsymbol{f}^b = \sum_{\tau=1}^{C} \tilde{\boldsymbol{H}}_{\tau:}^b (\hat{\boldsymbol{K}}^b \odot \boldsymbol{\Omega}_{C::}^b)_{\tau:} = \texttt{einsum("C m, C m -> m", } \tilde{\boldsymbol{H}}^b, (\hat{\boldsymbol{K}}^b \odot \boldsymbol{\Omega}_{C::}^b))$$

$$\boldsymbol{F}^b = \texttt{einsum("C Ci m, Ci m -> C m", } \boldsymbol{\Omega}^b, (\tilde{\boldsymbol{H}}^b \odot \hat{\boldsymbol{K}}^b))).$$

Here, $\boldsymbol{P}^b \in \mathbb{R}^{C \times C}$ is a lower triangular matrix with $\boldsymbol{P}_{ij}^b = \sum_{k=1}^{m} \boldsymbol{Q}_{ik}^b \hat{\boldsymbol{K}}_{jk}^b \boldsymbol{\Omega}_{ijk}^b$. A `matmul`-optimal computation for $\boldsymbol{P}^b$ is presented in Zhang et al. (2024) using a sub-tiling technique. However, a closer inspection of (12) reveals that the recurrence can be simplified by linearizing only the $\hat{\boldsymbol{h}}_t$ term, leading to:

$$\mathbf{S}_t = \hat{\mathbf{G}}_t \odot \mathbf{S}_{t-1} - \boldsymbol{h}_t \hat{\boldsymbol{k}}_t^\top, \quad \text{where } \hat{\mathbf{G}}_t = \mathbf{1}(\mu_t \beta_t + \tilde{\boldsymbol{h}}_t \odot \hat{\boldsymbol{k}}_t)^\top. \tag{25}$$

Here, $\hat{\mathbf{G}}_t$ is parameterized as a rank-one outer product, and hence this intra-chunk update can be computed efficiently using the parallel form of gated linear attention (GLA) presented in Zhang et al. (2024).

**Computing $\beta$.** For the normalization factor, $\beta_\tau$, in equation 10, we need to compute $\|\Delta \boldsymbol{s}_{i,\tau}\|^2 \; \forall 0 < i \leq m, \; 0 < \tau \leq C$ which require materializing $\Delta \boldsymbol{S}_\tau \in \mathbb{R}^{d \times m} \; \forall 0 < \tau \leq C$, that is memory and compute inefficient. However, we can see that these norms can be efficiently computed using matrix multiplication:

$$\Delta \boldsymbol{S}_\tau = -\gamma_t \nabla_S \mathcal{L}(\mathbf{S}_{t-1}, \boldsymbol{v}_t, \boldsymbol{k}_t) = \mathbf{1}(\tilde{\boldsymbol{h}}_t \odot \hat{\boldsymbol{k}}_t)^\top \odot \mathbf{S}_0 - \boldsymbol{h}_t \hat{\boldsymbol{k}}_t^\top$$

$$\boldsymbol{\beta}_\tau = \left(1 + \text{diag}[\Delta \boldsymbol{S}_\tau^\top \Delta \boldsymbol{S}_\tau]\right)^{-\frac{1}{2}}$$

$$\boldsymbol{d}_{S_\tau} := \text{diag}[\Delta \boldsymbol{S}_\tau^\top \Delta \boldsymbol{S}_\tau] = \text{diag}[\boldsymbol{S}_0^\top \boldsymbol{S}_0] \odot (\tilde{\boldsymbol{h}}_t \odot \hat{\boldsymbol{k}}_t)^2 + \|\boldsymbol{h}_t\|^2 \hat{\boldsymbol{k}}_t^2$$

We can also compute $\boldsymbol{d}_{S_\tau} \; \forall 0 < \tau \leq C$ for the entire chunk by matrix multiplications as

$$\boldsymbol{D}_S := [\boldsymbol{d}_{S_0}, \dots, \boldsymbol{d}_{S_C}] = \text{diag}[\boldsymbol{S}_0^\top \boldsymbol{S}_0] \odot (\tilde{\boldsymbol{H}} \odot \hat{\boldsymbol{K}})^2 + \text{diag}[\boldsymbol{H} \boldsymbol{H}^\top] \hat{\boldsymbol{K}}^2 \in \mathbb{R}^{C \times m}$$

$$[\boldsymbol{\beta}_0, \dots, \boldsymbol{\beta}_C] = (1 + \boldsymbol{D}_S)^{-\frac{1}{2}} \tag{26}$$

where all the square and square roots are performed element-wise.

This matrix form computes the states only at the end of each block massively save computation and I/O overhead, while enabling efficient use of the `matmul` operations on modern accelerators (Hua et al., 2022; Kacham et al., 2024; Dao & Gu, 2024; Zhang et al., 2024; Sun et al., 2023).

### B.3 PROOFS

#### B.3.1 ENCODER LAYER WITH STATE NORMALIZATION

The reconstruction loss in this case is

$$\mathcal{L}_t = \left\| \phi(\mathbf{S}_t)^\top \boldsymbol{v}_t - \boldsymbol{k}_t \right\|^2,$$

where $\mathbf{S}_t \in \mathbb{R}^{d \times m}$ with columns $\boldsymbol{s}_i$ (for $i = 1, \ldots, m$), $\boldsymbol{v}_t \in \mathbb{R}^d$, $\boldsymbol{k}_t \in \mathbb{R}^m$ and $\phi(\mathbf{S}_t) = [\phi_1, \ldots, \phi_m]$ is obtained by normalizing each column of $\mathbf{S}_t$; that is, $\phi_i = \frac{\boldsymbol{s}_i}{\|\boldsymbol{s}_i\|}$. Decomposing the reconstruction error in a per-basis (per-column) form and defining the reconstruction error,

$$\boldsymbol{e}_i := \phi_i^\top \boldsymbol{v}_t - (\boldsymbol{k}_t)_i \ \forall \, i = 1, \ldots, m$$

Then the loss is $\mathcal{L}_t = \sum_{i=1}^m \boldsymbol{e}_i^2$. By this decomposition, we can derive the gradient with respect to each column $\boldsymbol{s}_i$ separately:

$$\frac{\partial \mathcal{L}_t}{\partial \boldsymbol{s}_i} = 2 \, \boldsymbol{e}_i \, \frac{\partial \boldsymbol{e}_i}{\partial \boldsymbol{s}_i}.$$

The Jacobian of the normalized vector $\phi_i = \frac{\boldsymbol{s}_i}{\|\boldsymbol{s}_i\|}$, is

$$\nabla_{\boldsymbol{s}_i} \phi_i = \frac{1}{\|\boldsymbol{s}_i\|} \left( \mathbf{I}_d - \phi_i \phi_i^\top \right).$$

Thus, by the chain rule,

$$\frac{\partial (\phi_i^\top \boldsymbol{v}_t)}{\partial \boldsymbol{s}_i} = \frac{\partial \phi_i}{\partial \boldsymbol{s}_i} \, \boldsymbol{v}_t = \frac{1}{\|\boldsymbol{s}_i\|} \left( \mathbf{I}_d - \phi_i \phi_i^\top \right) \boldsymbol{v}_t.$$

Therefore, for each $i$ the gradient with respect to $\boldsymbol{s}_i$ is

$$\frac{\partial \mathcal{L}_t}{\partial \boldsymbol{s}_i} = \frac{2 \, \boldsymbol{e}_i}{\|\boldsymbol{s}_i\|} \mathbf{P}(\boldsymbol{s}_i) \, \boldsymbol{v}_t = 2 \, \boldsymbol{e}_i \, \frac{1}{\|\boldsymbol{s}_i\|} \left( \boldsymbol{v}_t - \frac{\boldsymbol{s}_i \, (\boldsymbol{s}_i^\top \boldsymbol{v}_t)}{\|\boldsymbol{s}_i\|^2} \right), \quad \text{for } i = 1, \ldots, m.$$

Here, the matrix $\mathbf{P}(\boldsymbol{s}_i) = \mathbf{P}(\phi_i) := \left( \mathbf{I} - \frac{\boldsymbol{s}_t \boldsymbol{s}_t^\top}{\|\boldsymbol{s}_i\|^2} \right)$ is known as the *projection matrix onto the orthogonal complement* of $\boldsymbol{s}_i$ in linear algebra (Strang, 2000, §3.3). Stacking these column gradients into the gradient with respect to the matrix $\mathbf{S}$ we obtain

$$\nabla_{\mathbf{S}} \mathcal{L}_t = \left[ \frac{\partial \mathcal{L}_t}{\partial \boldsymbol{s}_1}, \frac{\partial \mathcal{L}_t}{\partial \boldsymbol{s}_2}, \ldots, \frac{\partial \mathcal{L}_t}{\partial \boldsymbol{s}_m} \right],$$

Thus, the closed-form gradient can be expressed as:

$$\nabla_{\mathbf{S}} \mathcal{L}_t = \left[ \frac{2 \left( \phi_1^\top \boldsymbol{v}_t - (\boldsymbol{k}_t)_1 \right)}{\|\boldsymbol{s}_1\|} \left( \mathbf{I}_d - \phi_1 \phi_1^\top \right) \boldsymbol{v}_t, \ \cdots, \ \frac{2 \left( \phi_m^\top \boldsymbol{v}_t - (\boldsymbol{k}_t)_m \right)}{\|\boldsymbol{s}_m\|} \left( \mathbf{I}_d - \phi_m \phi_m^\top \right) \boldsymbol{v}_t \right] \tag{27}$$

$$= \left[ \frac{2 \, \boldsymbol{e}_1}{\|\boldsymbol{s}_1\|} \mathbf{P}(\boldsymbol{s}_1) \, \boldsymbol{v}_t, \ \cdots, \ \frac{2 \, \boldsymbol{e}_m}{\|\boldsymbol{s}_m\|} \mathbf{P}(\boldsymbol{s}_m) \, \boldsymbol{v}_t \right] \qquad \square$$

**Proposition B.1 (Proof of Proposition 3.1)** *Consider the unit sphere $\mathcal{C} = \{ \boldsymbol{s} \in \mathbb{R}^d \mid \|\boldsymbol{s}\| = 1 \}$ which is a which is a smooth Riemannian manifold. Let $\nabla_s \ell(\mathbf{s})$, the gradient of the loss $\ell$, and let $\nabla_{\mathcal{C}} \ell(\mathbf{s})$ be its orthogonal projection from the ambient space $\mathbb{R}^d$ onto the tangent space of the manifold at $\boldsymbol{s}$, denoted by $\mathcal{T}_{\mathcal{C}}(\boldsymbol{s})$.*

*In our update rule, the gradient term $\Delta \boldsymbol{s} = \alpha \, \mathbf{h}^{\perp \boldsymbol{s}}$ is constructed such that it is orthogonal to $\boldsymbol{s}$; that is, $\Delta \boldsymbol{s} \in \mathcal{T}_{\mathcal{C}}(\boldsymbol{s})$. Hence, we have*

$$\nabla_{\mathcal{C}} \ell(\mathbf{s}) = \nabla_s \ell(\mathbf{s})$$

*The gradient descent update of $\ell$ on the Riemannian manifold $\mathcal{C}$ is given by*

$$\mathbf{s}_{new} = exp_\mathbf{s} \left( -\eta_t \, \nabla_{\mathcal{C}} \ell(\mathbf{s}) \right),$$

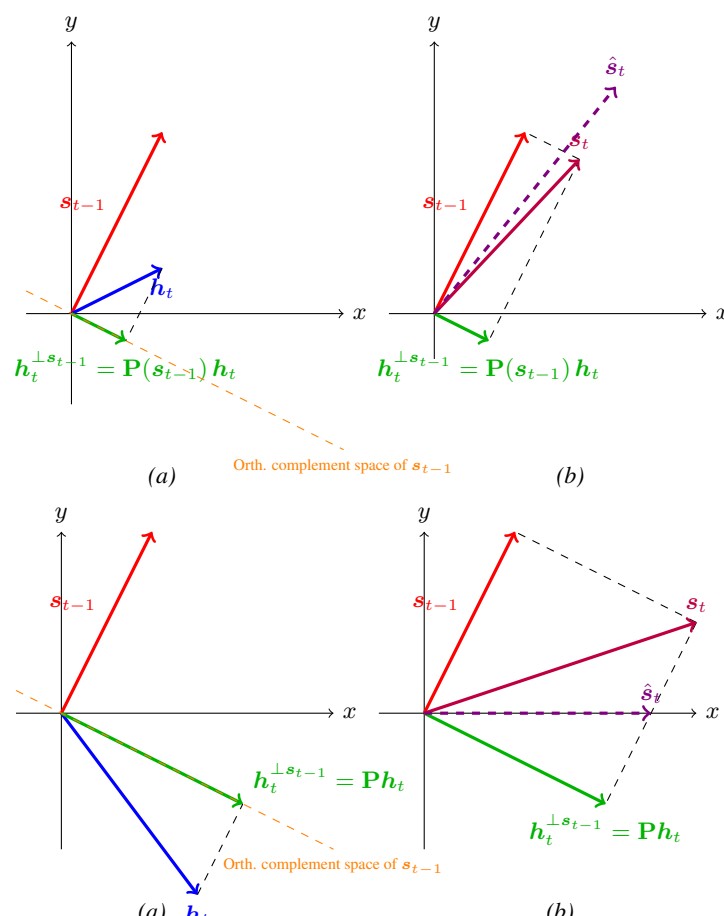

Figure 3: An illustration of the proposed update rule. (a) Example of a single memory slot state, $s_t$, an incoming token representation, $h_t$, and its component orthogonal to the current state, $h_t^{\perp s_{t-1}}$. (b) The updated state according to the proposed update rule, $s_t = s_{t-1} + \alpha_{i,t}\, h_t^{\perp s_{t-1}}$ contrasted with the updated state resulting from the superposition recurrence update used in standard linear attention: $\hat{s}_t = s_{t-1} + \alpha_{i,t}\, h_t$, (dashed arrow). A unit writing intensity ($\alpha_{i,t} = 1$) is assumed for simplicity in both recurrent update rules.

*where $\exp_{\mathbf{s}}$ is the exponential map on the sphere $\mathcal{C}$ (Absil et al., 2009; Boumal, 2023).*

*Replacing the exponential map with its first-order approximation, called retraction step, which projects from the tangent space onto the sphere manifold (Bonnabel, 2013). Therefore, our projected gradient update of the form $s_{i,t} = \mathcal{P}_{\mathcal{C}}(s_{i,t-1} + \Delta s_{i,t})$ (equation 10) is equivalent to performing a Riemannian gradient descent step with retraction on the manifold $\mathcal{C}$. $\square$*

> This proposition formalizes that by updating the memory slot with only the orthogonal component and then projecting back onto the unit sphere, we are effectively performing gradient descent on the Riemannian manifold of unit-norm state vectors.

## C  EXPERIMENT DETAILS

**Datasets:** We trained models on different datasets. For language modeling and common-sense reasoning tasks, models are trained on FineWeb-Edu dataset (Penedo et al., 2024) with context length of 4k. For sequence length scaling pattern models are trained on The Pile and Books3 dataset. The Pile is a large-scale, diverse corpus widely used for training and evaluating language models (Gao et al., 2020). It consists of a mixture of high-quality text sources, including books, academic papers, web content, and technical documentation. While it contains relatively few sequences exceeding

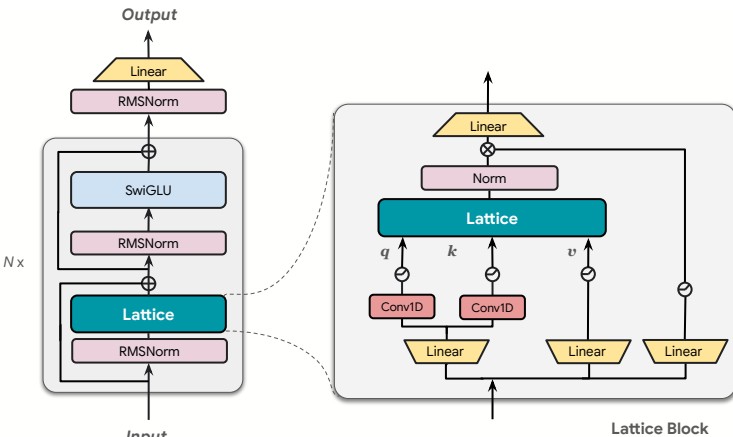

Figure 4: *(Left)* Block diagram of the language model. *(Right)* The Lattice block. Following the architecture used in Mamba (Gu & Dao, 2023), each sequence mixing block is composed of a pair of short `Conv1D` for the pair $\{q, k\}$ and the Lattice is followed by a `GeLU` post-gate.

$8k$ tokens, in this study, we restrict The Pile to a short-context setting with sequence lengths of 2k or 8k tokens. Books3, on the other hand, is a subset of The Pile that consists of high-quality, full-length books, commonly used for training language models for long-context evaluations. In the experiments we used this dataset to test model performance on sequences ranging from 512 to 16k tokens (in increments of $2\times$ per experiment). The same training setup as The Pile is applied to ensure consistency. Since Books3 contains structured narratives and long-form content, it provides a rigorous test of a model's ability to track dependencies over extended contexts.

For all experiments, the training batch size is fixed at 0.5 million tokens, irrespective of sequence length. This means that for a given context length $T$, each batch contains $0.5M/T$ sequences.

**Baseline Models and Model Architecture** We compare our method against Transformer++ model (Touvron et al., 2023) as well as the following sub-quadratic sequence models: Linear-Attention (LA) (Katharopoulos et al., 2020), TTT (Sun et al., 2024), DeltaNet (Yang et al., 2024c), Gated DeltaNet (Yang et al., 2024a), Mamba2 Dao & Gu (2024). As discussed in the paper, the Lattice layers incorporate $\ell_2$-normalization on the state, whereas TTT applies $\ell_2$-normalization on the output of the decoding layer.

**Model Architecture** For sub-quadratic sequence models, we adopt the architectural setup used in Mamba (Gu & Dao, 2023), where each sequence-mixing block consists of a pair of short `Conv1D` layers for the $\{q, k\}$ pair, which share a linear projection. A `GeLU` post-gate is applied to the output of the sequence model. The Transformer++ model, on the other hand, follows the architecture proposed in LLaMA (Touvron et al., 2023). All the models follow the multi-head structure introduced in Transformers (Vaswani et al., 2017a). The model architecture used for Lattice is illustrated in Figure 4.

Instead of using a fixed learning rate as in standard gradient descent, we model the learning rate of the compression layer as an input-dependent neural network, $\gamma_t = \text{sigmoid}(\mathbf{W}_\gamma \boldsymbol{x}_t)$. This is a key benefit of the bilevel optimization setup, as the weights of this network can be trained in the outer loop along with the rest of the model weights.

**Hyperparameters.** The block diagram of the language model is in Figure 4, and the model hyperparameters are listed in Table 4. We use the AdamW optimizer with a cosine learning rate schedule, which includes a warm-up phase of 1k steps (0.5 billion tokens) and a final learning rate of 3e-5. We apply a weight decay of 0.1 and gradient clipping at 1.0. The width of the short convolution layer is set to 4. For language model training on FineWeb-Edu and the reasoning tasks in Table 2, a default chunk size of $C = 4$ is used for Lattice unless otherwise specified. For the ablation studies with small models in Table 3 and the long-context scaling experiments (110M models, Books

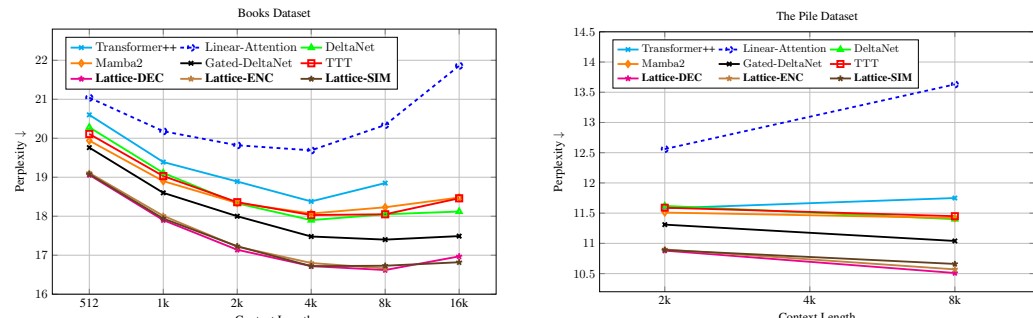

Figure 5: Model perplexity as a function of context length for models of size 110M parameters. *(Left)* displays results for the Books dataset vs context length $\{512, 1024, 2k, 4k, 8k, 16k\}$ ; *(Right)* shows results for The Pile dataset vs context length $\{2k, 8k\}$. Note that pre-training Transformers from scratch often performs poorly on very long contexts (e.g., 16k); the common approach is finetuning from shorter-context models (Touvron et al., 2023). Therefore, the Transformer results shown here are limited to context lengths $T \leq 8k$.

dataset) a chunk size of $C = 1$ is used to evaluate the exact, non-approximated performance of the compression layer.

| Configuration | $n_{\mathbf{blocks}}$ | $d_{\mathbf{model}}$ | $d_{\mathbf{head}}$ | Peak learning rate |
|---|---|---|---|---|
| 110M params / 5B tokens | 12 | 768 | 64 | 1e-2 |
| 340M params / 15B tokens | 24 | 1024 | 64 | 1.5e-3 |
| 760M params / 30B tokens | 24 | 1536 | 64 | 1.25e-3 |

Table 4: Model and training hyperparameters.

**Ablation: Scaling Memory Size.** Figure 6 illustrates the scaling behavior of Lattice with respect to the number of memory slots $m$. We observe that the performance of Lattice scales linearly with respect to $\log(m)$ (noting the logarithmic scale on the x-axis). Notably, Lattice with only a quarter of the memory slots ($m = d/4 = 16$) outperforms TTT with full memory ($m = 64$), which empirically validates that the proposed orthogonal update mechanism utilizes the fixed-state state capacity significantly more efficiently, enabling Lattice to achieve $4\times$ compression gain. It is worth noting that while increasing $m$ yields consistent gains, models with $m > d$ come at the cost of increased parameter counts in the embedding projection matrices $\mathbf{W}_q$ and $\mathbf{W}_k$ (and vice versa for models with $m < d$).

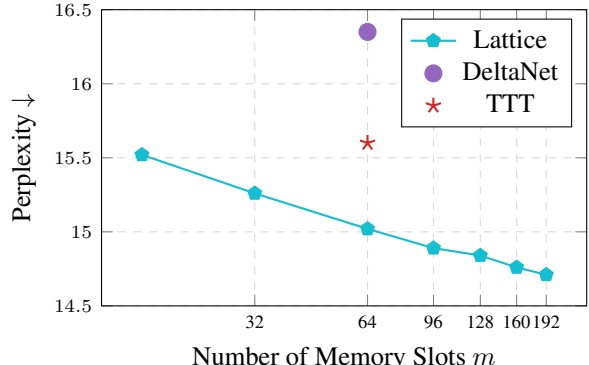

Figure 6: **Ablation on Memory Size ($m$).** Validation perplexity of Lattice's (110M parameters), trained on FineWeb-Edu with varying number of memory slots $m$. For comparison, DeltaNet and TTT are plotted at their standard configuration of $m = 64$ (equivalent to head dimension $d = 64$). Lattice consistently outperforms baselines, achieving lower perplexity even with half the memory capacity.

Table 5: Ablation study of Lattice's per-token vs. per-chunk normalization (110M parameters, trained on FineWeb-Edu). The perplexity of the chunk-wise approximation (13) using per-token vs. per-chunk normalization strategies, where the normalization step is only applied after each chunk.

| Configuration | ppl ↓ |
|---|---|
| DeltaNet | 16.35 |
| TTT C＝1 | 15.60 |
| Lattice C＝4 Per-token | 15.15 |
| Lattice C＝4 Per-chunk | 15.17 |
| Lattice C＝16 Per-token | 15.32 |
| Lattice C＝16 Per-chunk | 15.36 |

**Ablation: Per-token vs. Per-chunk Normalization.** Table 5 compares the perplexity of the chunk-wise approximation (13) using per-token versus per-chunk normalization strategies, where the latter applies normalization solely at chunk boundaries. The results indicate that the performance divergence between the two strategies widens as the chunk size increases. Furthermore, for the simplified approximation (15) with a chunk size of $C = 16$, we observed that per-chunk normalization resulted in training instability. These results highlight the importance of per-token normalization for ensuring both performance and stability with larger chunk sizes. We attribute this to the fact that applying normalization exclusively at chunk boundaries creates a discrepancy between intra-chunk and inter-chunk update dynamics.

## C.1 RECALL-INTENSIVE TASK: MULTI-QUERY ASSOCIATIVE RECALL

To rigorously evaluate the memory capacity and in-context learning capabilities of Lattice, we evaluate the model on Multi-Query Associative Recall (MQAR) task introduced by Arora et al. (2023).

Prior research indicates that a model's capability in associative recall task is strongly predictive of its in-context learning quality and language modeling performance (Olsson et al., 2022). While standard associative recall tests a model's ability to retrieve a value for a single query after a sequence of key-value pairs, MQAR introduced by Arora et al. (2023) as a more challenging and realistic scenario where the model must perform multiple recalls at varying positions within a single forward pass. In this task, a sequence consists of a series of key-value pairs followed by a series of queries (keys) mixed with other tokens. The model must correctly predict the value associated with a specific key. An example sequence is formatted as follows:

$$\text{A 4 B 3 C 6 F 1 E 2} \ldots \text{A ? C ? F ? E ? B ?} \rightarrow \text{4 6 1 2 3}$$

In this setup, the model must compress the association of the key-value pairs in its memory state over potentially long distances and retrieve them correctly when queried multiple times. Therefore, MQAR serves as a strong benchmark for the memory capacity of recurrent architectures while Transformer architecture can easily solve this task.

**Experimental Setup.** We follow Arora et al. (2023) and adopt the experimental setup used in Beck et al. (2024) for the more difficult setting. We generate 100,000 synthetic training samples and 3,000 validation samples from a vocabulary size of 8192. All models are trained for 64 epochs using the AdamW optimizer with a cosine annealing learning rate schedule after a 10% linear warm-up phase and with a weight decay of 0.1.

We train models with two blocks using a single head, while varying the Model Dimension ($d$) to observe scaling behaviors. We set the number of slots $m = d$ for Lattice, TTT, DeltaNet, and GLA; consequently, each block maintains a state matrix of size $d^2$. In comparison, xLSTM uses an additional normalizer state vector, resulting in a larger state size of $d^2 + d$ per block.

We evaluate Lattice on two challenging settings across different context lengths ($L$) and high numbers of key-value pairs. For each setting, we sweep over learning rate grids to ensure optimal convergence:

- Context Length $L$=1024: We sweep learning rates over $\{$1e-2, 3.16e-3, 1e-3, 3.16e-4, 1e-4$\}$ with a batch size=96.

- Context Length $L$=2048: We sweep learning rates over $\{$1e-2, 1e-3, 2.2e-4, 5e-5, 1e-5$\}$ with a batch size=24.

For each context length, we scale the difficulty by varying the number of Key-Value pairs that must be memorized, testing $N_{KV} \in \{48, 96, 256\}$. This progression specifically evaluates memory capacity of the model, determining whether the architecture can effectively store and recall a high density of information within the fixed-size state, thereby testing the efficiency of the compression mechanism proposed by Lattice.

As the results in Figure 7 show, Lattice offers the best MQAR accuracy over the majority of the settings and it exhibits a significant improvement over its main counterparts: TTT and DeltaNet. Furthermore, while baselines exhibit performance degradation as context length increases, Lattice maintains its accuracy across both context lengths. *This overall shows the efficient compression mechanism of Lattice, demonstrating its superior capability to store and retrieve dense information within its fixed-size recurrent memory.*

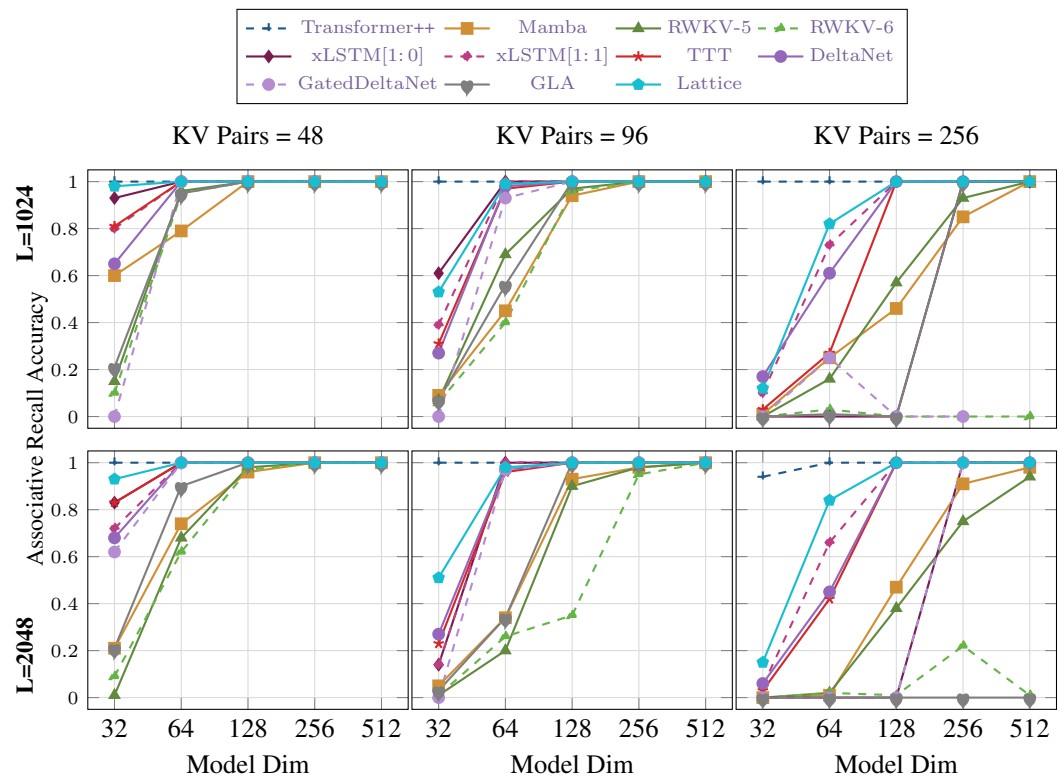

Figure 7: Associative Recall Accuracy (%) on the recall-intensive MQAR task. The baseline results for Transformer, Mamba (Gu & Dao, 2023), RWKV-5/6 (Peng et al., 2023; 2024), xLSTM[1:0], and xLSTM[1:1] are taken from (Beck et al., 2024), while we trained Lattice, TTT, DeltaNet, GatedDeltaNet, and GLA. Notably, GatedDeltaNet and GLA were unable to effectively learn this task for the 256 Key-Value pair setting across most model dimensions.

## C.2 THROUGHPUT COMPARISON.

The training throughput are compared in Figure 8. For Transformer++, we use Flash-Attention-2 (Dao, 2023) with a block size of 256, while the rest of the models use a chunk size of 64. The results demonstrate that Lattice exhibits superior throughput scalability compared to Transformers as sequence length increases, while remaining competitive with TTT. Our results confirm that Lattice maintains the favorable sub-quadratic scaling characteristic of RNNs while offering improved

expressivity. Both Lattice and TTT are implemented in pure Jax, without relying on specialized hardware-optimized kernels.

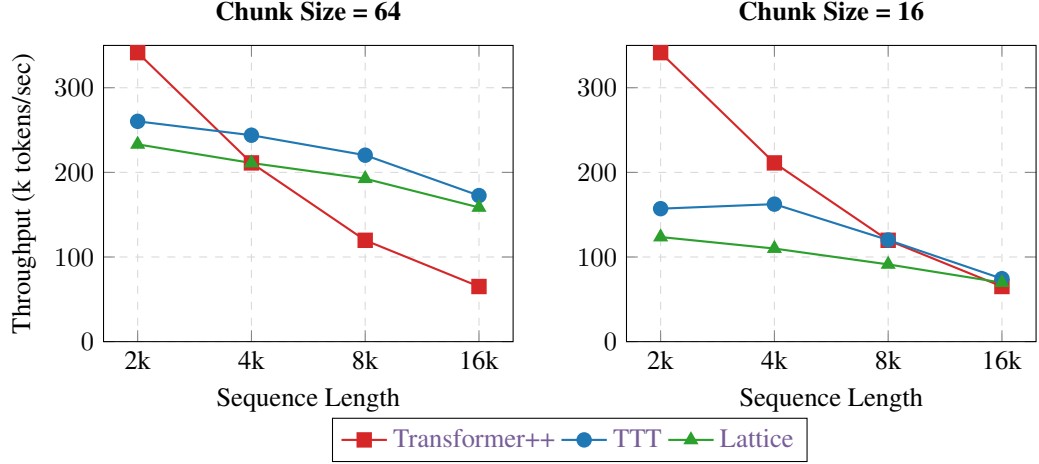

Figure 8: **Training Throughput vs. Sequence Length.** Throughput (in k tokens/sec) for 110M parameter models trained on a 2×2 TPU Trillium setup. **Left:** Throughput using a Chunk Size of 64. **Right:** Throughput using a Chunk Size of 16. Due to its recurrent formulation, Lattice maintains higher throughput than Transformer++ as sequence length increases, remaining competitive with TTT. Lower chunk sizes generally reduce throughput due to lower FLOP utilization and reduced parallelism.

## D   IMPLEMENTATION

```python
from functools import partial
import jax
import jax.numpy as jnp
from flax import linen as nn
from jax import lax, vmap
from einops import rearrange
from jax.numpy.linalg import norm

@partial(jax.jit, static_argnames=['mode', 'chunk_size'])
def lattice_compress(S0, K, V, Q, eta, mu, mode='Dec', chunk_size=16):
    """
    A simple Lattice operation with recursive update.
    S0: (BxHxMxD)
    K: (BxLxHxM), V: (BxLxHxD), Q: (BxLxHxM), eta/mu: (BxLxHx1)
    """
    # split chunks and transpose to B H NC C d
    K = rearrange(K, 'b (nc c) h m -> b h nc c m', c=chunk_size)
    V = rearrange(V, 'b (nc c) h d -> b h nc c d', c=chunk_size)
    Q = rearrange(Q, 'b (nc c) h m -> b h nc c m', c=chunk_size)
    eta = rearrange(eta, 'b (nc c) h 1 -> b h nc c 1', c=chunk_size)
    mu = rearrange(mu, 'b (nc c) h 1 -> b h nc c 1', c=chunk_size)

    if chunk_size == 1:  # scan over sequence
      scan_fn = partial(compress_step, mode=mode)
    else:  # scan over chunks
      scan_fn = partial(compress_chunk, mode=mode)

    def compress_scan(s0, k,v,q,e,m):
        return lax.scan(scan_fn, s0, (k,v,q,e,m))

    # vmap over batch and heads
    batch_head_scan_fn = vmap(
        vmap(compress_scan, axis_name="heads"),
        axis_name="batch")
    S, Y = batch_head_scan_fn(S0, K, V, Q, eta, mu)
    Y = rearrange(Y, 'b h nc c d -> b (nc c) h d')
    return S, Y

def compress_step(S, inputs_t, mode='Dec'):
  # S: MxD
```

```
42   # inputs_t: k:1xM, v:1xD, q:1xM, eta:1x1, mu:1x1
43   k, v, q, eta, mu = inputs_t
44
45   s_norm = norm(S, axis=-1, keepdims=True) + 1e-6
46   Phi = S / s_norm
47
48   if mode == 'Dec':
49     h = k @ Phi - v
50   elif mode == 'Sim':
51     h = - v
52
53   # 1) orthogonal projection  h_t^⊥s_i (8), (9) : compute Delta    Equation 12
54   h_hat= (h @ Phi.T) / s_norm.T
55   k_hat = k / s_norm.T
56   Delta1 = k_hat.T @ h
57   Delta2 = S * (h_hat * k_hat).T
58   Delta = Delta1 - Delta2
59
60   # #VERIFY ORTHOGONALITY of Delta and S
61   # jax.debug.print("<Delta,S>: {}",jnp.einsum('md, md->m',Delta,S))
62
63   # 2) update S and normalize it  β_t
64   S_t = mu * S - eta * Delta
65   beta = s_norm * jax.lax.rsqrt(
66       ((mu*S)**2 + (eta * Delta)**2).sum(axis=-1, keepdims=True))
67   # beta = s_norm / (norm(S_t, axis=-1, keepdims=True)+ 1e-6)
68   S_t *= beta       #  (Equation 10)
69   # readout
70   y = q @ S_t
71   return S_t, y
72
73 def compress_chunk(S, inputs_t, mode='Dec'):
74   # S: MxD
75   # inputs_t: K:CxM, V:CxD, Q:CxM, eta:Cx1, mu:Cx1
76   K, V, Q, eta, mu = inputs_t
77
78   s_norm = norm(S, axis=-1, keepdims=True) + 1e-6 # Mx1
79   Phi = S / s_norm # MxD
80
81   if mode == 'Dec':
82     H = K @ Phi - V  # CxD
83   elif mode == 'Sim':
84     H = - V  # CxD
85   H_hat= (H @ Phi.T) / s_norm.T  # CxM :  Equation 12
86   K_hat = K / s_norm.T  # CxM
87
88   # compute  ‖ΔS‖² and β Equation 26
89   Delta1_sq = K_hat**2 * (H*H).sum(axis=-1)[:, None] # CxM
90   Delta2_sq = (s_norm.T**2) * (H_hat * K_hat)**2      # CxM
91   Delta_sq = jnp.clip(Delta1_sq - Delta2_sq, min=0.) # CxM
92   Delta_sq = eta**2 * Delta_sq
93   beta = s_norm.T * jax.lax.rsqrt(eta**2 * s_norm.T**2 + Delta_sq)    # (CxM)
94
95   G_gate = beta * (mu+ eta*(H_hat*K_hat)) # CxM :  Equation 15
96
97   # Parallel intra-chunk based on linear attention
98   # with vectorized forgetting gate (GLA):  Equation 15
99   K_tilde = -beta * eta * K_hat   # CxM
100  S_t, Y = GLA_intra_chunk(Q=Q, K=K_tilde, V=H, G=G_gate, S0=S)
101  return S_t, Y
```

Listing 1: A simple implementation of the Lattice update rule.

**Limitation and Future Work.**    The high expressiveness of the proposed orthogonal projection comes at the cost of an extra $\mathcal{O}(d\,m)$ computation compared to simpler linear RNNs, as detailed in section refsec:complexity. However, the final update rule, presented in its chunk-wise form in sections 3.4 and B.2.1, relies entirely on standard, hardware-efficient operators such as matrix and element-wise multiplications. While the proposed chunk-wise approximations in Equation 13 and Equation 15 enable parallelizable matrix multiplications within chunks, the non-linear nature of this test-time training prevents the use of inter-chunk parallelization via associative scans, which limits hardware utilization compared to fully linear recurrences. We believe that exploring hardware-efficient strategies remains a key area for future work. Specifically, tailoring techniques such as the large-chunk training strategies proposed in LaCT (Zhang et al., 2025), or the multi-stage and hierarchical process proposed in Li et al. (2025) for the non-linear recurrence of Lattice represents a

promising direction to maximize hardware utilization while retaining the benefits of the principled orthogonal update.

**LLM Usage.** We acknowledge the use of a large language model (LLM) solely for improving the linguistic quality and clarity of this manuscript and polishing the paper.

# E SUMMARY OF REVISIONS

1. **New Experimental Results**
   - **Memory Capacity Ablation (Figure 6):** We conducted a new ablation study scaling the memory size ($m$). Results show that Lattice with only 16 slots ($m = d/4$) outperforms the TTT baseline with 64 slots ($m = d$). This empirically validates the superior efficiency of the orthogonal update mechanism.
   - **Recall-Intensive Task (MQAR) (Figure 7):** We added the Multi-Query Associative Recall (MQAR) benchmark to evaluate memory capacity under stress. Lattice demonstrates significant improvements in recall accuracy compared to TTT and DeltaNet, particularly as sequence length increases, highlighting its robustness against memory interference.
   - **Training Throughput vs. Sequence Length Figure 8:** Throughput for 110M parameter models trained using a chunk Size of 64 and 16.
   - **Per-token vs. Per-chunk Normalization Ablation (Table 5):** We compared the perplexity of the chunk-wise approximation using per-token versus per-chunk normalization strategies. These results highlight the importance of per-token normalization for ensuring both performance and stability with larger chunk sizes.

2. **Theoretical Clarifications and Comparisons:**
   - **Relationship with LaCT and Muon (Discussion Section):** We added a detailed discussion distinguishing Lattice from the concurrent work LaCT (Test-Time Training Done Right).
   - **Comparison with RWKV:** We expanded Table 1 and the results section to include comparisons with the RWKV family (RWKV-6, RWKV-7), ensuring a more comprehensive evaluation against state-of-the-art linear RNNs. Also included performance comparison to RWKV-5/6 in Figure 7.
   - **Throughput Comparison :** We clarified the throughput comparison on TPUs, where Lattice exhibits linear scalability with respect to sequence length, significantly outperforming Transformers on long contexts (e.g., 16k tokens).

3. **Implementation** We included a JAX/Flax pseudocode implementation of the Lattice update rule (both step-wise and parallel chunk-wise).

4. **Manuscript Improvements**: We updated **Limitation and Future Work** paragraph and corrected the typos.

