# OpenReview forum: "Lattice: Learning to Efficiently Compress the Memory"
_ICLR.cc/2026/Conference — Submitted to ICLR 2026_

### Official Review · Reviewer_ViSx · 2025-10-24

**Soundness:** 3
**Presentation:** 3
**Contribution:** 4
**Rating:** 6
**Confidence:** 2

**Summary:**

This paper introduces Lattice, a novel recurrent neural network (RNN) mechanism designed to address the quadratic computational and memory complexity of attention mechanisms in sequence modeling. Lattice leverages the low-rank structure of key-value (K-V) matrices to compress the memory cache into a fixed number of slots, achieving sub-quadratic complexity. The memory update is formulated as an online optimization problem, leading to a dynamic, interpretable, and state/input-dependent update rule. The core innovation is an orthogonal update: each memory slot is updated only with information orthogonal to its current state, thus minimizing redundancy and interference. Experimental results show that Lattice outperforms strong baselines (including Transformer++, Mamba2, DeltaNet, TTT, and others) in perplexity and zero-shot reasoning tasks across various context lengths and model sizes.

**Strengths:**

* The paper proposes a principled, optimization-based approach to memory compression in sequence models, moving beyond heuristic or ad-hoc memory management.
* The orthogonal update rule is mathematically elegant and well-motivated, with clear connections to Riemannian optimization and online dictionary learning.
* The update mechanism is highly interpretable: each memory slot only absorbs new, non-redundant information, which is a desirable property for long-context modeling.

**Weaknesses:**

* While ablations are provided, the analysis could be deepened. For example, the impact of the number of memory slots (m), or the sensitivity to the choice of normalization and regularization, could be explored more systematically.
* Although the method is theoretically efficient, the actual training/inference wall-clock time and memory usage compared to baselines (especially on very long sequences) are not deeply analyzed.

**Questions:**

* How sensitive is Lattice’s performance to the number of memory slots (m)? Is there a clear trade-off between slot count, computational cost, and model expressivity?
* How does the model utilize the memory slots in practice? Are some slots consistently more active than others? Can you visualize or analyze the diversity of information stored in different slots?
* Can you provide a pytorch-like pseudo code to help clarify the algorithm and facilitate implementation?

---

> ### Author Response · Authors · 2025-11-24
>
> We thank the reviewer for their constructive feedback and for recognizing the principled nature, mathematical elegance, and interpretability of our approach.  Below, we address the suggestions/concerns:
>
> - **Sensitivity to Memory Slots ($m$) & How the model utilizes the memory slots in practice:** Thanks for your suggestion, we have conducted a new ablation study to systematically analyze this sensitivity. As illustrated in the new ablation study (included in Table 3 and Appendix C), Lattice's performance scales linearly with $\log(m)$. Remarkably, *\model with only a quarter of memory slots ($m=16$) outperforms TTT with full memory ($m=64$)*. This empirically validates that *the proposed orthogonal update mechanism utilizes the fixed-state state capacity significantly more efficiently*, allowing Lattice to achieve $4\times$ compression gain. While increasing $m$ improves perplexity, models with $m > d$ incur increased parameter costs in the projection matrices (and vice versa for models with $m < d$).
>
> - **Training Wall-clock Time:** In Figure 6, we provide training throughput comparisons on TPUs. Lattice share a similar pattern as recurrent models in general for throughput scalability and exhibit superior throughput scalability compared to Transformers as sequence length increases.
>
> - **Code and  Parallelism:** We have included a detailed implementation and code in the Appendix of the revised paper. It explicitly maps the mathematical derivations (orthogonal projection, normalization) to standard parallelizable tensor operations (matmul, and $\odot$).

---

### Official Review · Reviewer_PU48 · 2025-11-01

**Soundness:** 2
**Presentation:** 2
**Contribution:** 3
**Rating:** 4
**Confidence:** 3

**Summary:**

This paper introduces Lattice, a novel sub-quadratic RNN architecture. The core innovation is the orthogonal memory update for mitigating memory interference with previously stored information by updating each memory slot exclusively with information orthogonal to its current state。The expriments analyze the proposed model on common-sense reasoning tasks and the effectiveness of each proposed component.

**Strengths:**

1. The idea of ​​orthogonal memory update is interesting and reasonable for alleviating memory conflict problems.
2. It provides a thorough theoretical analysis, starting with online gradient descent, to derive the form of orthogonal memory update, which has an elegant mathematical form and high interpretability.

**Weaknesses:**

1. The paper only reports common-sense reasoning tasks to evaluate language modeling capabilities, lacking experiments on recall-intensive and long-context tasks (such as MQAR[1] and LongBench[2], etc.), making it difficult to know the model's contextual understanding ability and thus difficult to fully evaluate the effectiveness of the proposed architecture.
2. Lacking open-source code and models, I cannot know the specific details and effectiveness of the proposed architecture implementation, or whether the implemented operators are truly hardware-efficient and parallel-trained.

[1] Simran Arora, et al. Zoology: Measuring and improving recall in efficient language models. ICLR, 2024.
[2] Yushi Bai, et al. Longbench: A bilingual, multitask benchmark for long context understanding. ACL, 2024.

**Questions:**

If each update of memory is orthogonal to the previous memory, does this mean there will be no interference from historical memories? How can we experimentally verify that this can alleviate memory conflict and thus improve model performance with the same memory/state capacity?

---

> ### Author Response · Authors · 2025-11-24
>
> We thank the reviewer for their constructive feedback and for appreciating the theoretical depth and elegance of the proposed design, particularly the orthogonal memory update. Below, we address the concerns raised by the reviewer:
>
> - **Recall-Intensive Tasks (MQAR):** In the revised manuscript (Appendix C), we have included results for the Multi-Query Associative Recall (MQAR) task under challenging settings (high number of key-value pairs and long sequences). Lattice demonstrates significant improvement in recall accuracy compared to baselines, particularly TTT. Especially, while baselines exhibit performance degradation as context length increases, Lattice maintains robust accuracy. This empirical study directly supports our claim that the orthogonal update mechanism effectively compresses and retrieves dense information over long contexts. Furthermore, we point to Figure 2 (Left), which shows perplexity scaling on the Books3 dataset up to 16k tokens. Lattice consistently outperforms baselines, and notably, the performance gap widens as the context length increases.
> - **Code and  Parallelism:** In the revised paper, we have included the code besides the core implementation details for the Lattice’s update rule. As evidenced by the implementation and Eqs. 13-16 (and Eq. 27)  in Section 3.4.1, our derived parallel chunk-wise form relies entirely on standard, hardware-efficient operators: matrix multiplications and element-wise multiplication ($\odot$).
>
> - **Question about Orthogonality and Memory/State Capacity:** The orthogonal update ($\Delta s_t \perp s_{t-1}$) ensures that the new information being written does not project onto (and thus overwrite) the existing state vector direction and it only stores the "residual" information not already represented in the basis. This orthogonality is structurally enforced and can be verified numerically by uncommenting the line 61 of the code snippet. Empirically, since MQAR requires retrieving specific values associated with specific keys after a long delay, the performance improvement of Lattice on MQAR directly supports that the orthogonal mechanism improves compression and alleviates memory conflicts better than standard updates. Moreover, ablation Study (Table 3): Lattice model significantly improves perplexity (15.60 $\to$ 15.07) compared to baselines without orthogonal updates (like TTT) for the *same memory size*. Also, a new ablation study on memory size scaling show that \model with only 16 memory slots ($m=d/4$) outperforms TTT with 64 slots (full memory), empirically proving that the orthogonal mechanism utilizes capacity significantly more efficiently than standard updates by minimizing redundant storage.

---

> > ### Comment · Reviewer_PU48 · 2025-11-26
> >
> > The authors' reply answered most of my questions, and I have increased my rating to 6.

---

### Official Review · Reviewer_Coa9 · 2025-11-03

**Soundness:** 3
**Presentation:** 3
**Contribution:** 2
**Rating:** 4
**Confidence:** 4

**Summary:**

See Below

**Strengths:**

1. comprehensive related work coverage: this paper covered almost all related like Transformer++ / GLA / Mamba2 / DeltaNet / TTT, especially the TTT, but missing RWKV

**Weaknesses:**

1. Novelty: The paper's core contribution is an incremental modification of TTT-Linear. Both methods use online optimization with reconstruction loss to update memory states. The main difference is that lattice applies normalization to state columns and adds orthogonal projection, while TTT-Linear normalizes the output. The orthogonal projection itself has been standard practice for rnn stabilization since unitary rnn.
2. The experimental results show minimal improvements over existing methods.

**Questions:**

From theoretical analysis, Lattice should have higher computational cost than Mamba2 because the orthogonal projection requires additional operations for each memory slot, and tensor contractions are more complex than simple outer products. However, the appendix claims Lattice shows "better throughput scalability" without providing concrete measurements. Given that Mamba2 has highly optimized CUDA kernels, the throughput claims appear contradictory.

---

> ### Author Response · Authors · 2025-11-24
>
> We thank the reviewer for their valuable feedback. Below, we clarify the novelty of our approach and address the questions.
>
> - **Novelty**
>   - **Comparison with with  TTT**:
> > "Both methods use online optimization with reconstruction loss to update memory states."
>   - While we share the high-level "online optimization" framework, reducing Lattice to an incremental modification of TTT overlooks the fundamental differences in the mechanism and its theoretical derivation.
>
>
>   - **Mechanism & Interpretation:** Lattice propose **Orthogonal State Recurrence (OSR)**: an interpretable input and state dependent update rule,  which involves projecting the input onto the orthogonal complement of the current state, enforcing "non-redundant" compression by adding only novel information.
>   -  The normalization step in Eq. 10 (after the orthogonal projection), is not merely a stability tweak; as discussed in Section 3.2,   it effectively induces a data- and state-dependent gating mechanism (forget and input gates). This stabilizes the update while managing memory capacity, offering an approach for designing **a novel and effective gating mechanisms** in RNNs, theoretically supported by Proposition 3.1 . The ablation study in Table 3 emphasizes the distinct importance of each of these steps (orthogonal recurrence and normalized projection) to the final performance.
>
>   - **Theoretical Derivation:** As summarized in Table 1, while most recurrent models (e.g., Mamba2, DeltaNet, RWKV) can be viewed as online learners of reconstruction loss, we provide two distinct perspectives for deriving Lattice's Normalized Orthogonal State Recurrence (NOSR):
>     - I) We formulate compression as online gradient descent on a decoding layer with normalized weights $\phi(S_t)$ , then apply normalization to prevent unbounded state growth (a consequence of the Pythagorean theorem ). This contrasts with TTT, which applies nonlinearity to the output activations and does not explicitly address state explosion/vanishing.
>     - II) In Proposition 3.1, we prove that NOSR can be derived by gradient descent on a Riemannian manifold with retraction applied on the linear decoding (regression) layer. This analytical approach provides a geometric grounding distinct from simply choosing a heuristic nonlinearity.
>
>
>   - **Parallelization:** Linearizing this specific recurrence for chunk-wise computation leads to a unique formulation (Eq. 13) , which requires the specialized parallelization strategy detailed in Section 3.4.1, distinct from the standard chunk-wise approach used in TTT.
>
>
> - **Empirical Significance:** The distinction is not just theoretical. In Table 3 (Ablation), we show that moving from a TTT-like baseline to the Orthogonal Update rule + Normalization improves perplexity significantly from 15.60 to 15.02. Furthermore, the results for context length scaling and model size scaling in Figure 2 , as well as zero-shot common-sense reasoning in Table 2, demonstrate significant performance improvements (~9% over TTT for L=16K).  Moreover, the new results for the Recall intensive task (Multi-Query Associative Recall (MQAR)) (added to appendix C) highlight how significantly Lattice's efficient compression mechanism enhances the model capability to store and retrieve dense information compared to baselines, particularly TTT. Finally, ablation studies demonstrate that Lattice with only a quarter of memory slots ($m=16$) outperforms TTT with $m=d=64$. These improvements confirm that the specific mechanism matters, not just the general online optimization framework.
>
>
> - **Comparison with Unitary RNNs**: We respectfully emphasize the distinction between our method and Unitary RNNs.
> Unitary RNN is a nonlinear recurrent model that constrain the recurrent weight matrix, ($W$),  that model latent  state  to latent state link,  in $h_{t+1} = \sigma( W h_{t} + \Delta_t)$,  to be unitary ($W^H W = I$) with the unique goal of preventing exploding/vanishing gradients (though due to the nonlinearity, the parametrization can not guarantee that gradients do not vanish).
> In contrast, Lattice applies **orthogonal projection to the update step** ($\Delta s$, the incoming information) and  projects it onto the orthogonal complement of the current state, ($h_t^{\perp s_{t-1}}$). This relative projection ensures non-redundant compression of information rather than just signal propagation stability.

---

> ### Author Response · Authors · 2025-11-24
>
> - **Throughput Comparison**
> The goal of the throughput analysis (Figure 6) was to demonstrate that Lattice, remains competitive with recurrent \method{TTT} for throughput scalability *compared to Transformers* as sequence length increases. As shown, these recurrent models scale better than Transformers as sequence length increases.
> Regarding the computational cost:
>    - *Algorithmic Complexity:* While Lattice involves projection matrices, Section 3.4 explicitly avoids full matrix-vector multiplication by exploiting the rank-one form of the projection $P$, reducing the complexity of this step to $\mathcal{O}(dm)$. As the added code in the appendix show these computations rely entirely on standard, hardware-efficient operators: matrix multiplications and element-wise multiplication ($\odot$).
>    - *Implementation Details:* We implemented Lattice in pure Jax, hence in throughput comparison we used  a similar Jax implementation for all methods to ensure a fair baseline, without relying on specialized kernels (e.g. using XLA). We acknowledge that models like Mamba2 benefit from highly optimized CUDA kernels and parallelization across chunks, leading to higher raw throughput.
> We explicitly discuss this trade-off in the Limitation and Future Work section, noting that the higher expressivity of Lattice comes with a computational overhead compared to linear RNNs .
>
> We have clarified these in the revised manuscript.
>
>
> - **RWKV Comparison**
> Per your suggestion, we have added a comparison with the RWKV family Table 1 and included the MQAR accuracy in Appendix C of the revised paper to ensure a more complete evaluation.

---

### Official Review · Reviewer_SnZw · 2025-11-03

**Soundness:** 3
**Presentation:** 2
**Contribution:** 2
**Rating:** 4
**Confidence:** 3

**Summary:**

This paper introduces Lattice, a novel RNN mechanism leveraging the low-rank structure of K-V matrices to compress cache into fixed memory slots, achieving sub-quadratic complexity via online optimization and gradient-based dynamic updates. Its core innovation is orthogonal updates that add only non-redundant information to each memory slot, minimizing interference. Experiments show Lattice outperforms baselines in perplexity across diverse context lengths, with more significant gains as contexts grow longer.

**Strengths:**

- The work addresses fundamental and forward-looking architectural challenges in sequence models, which are of significant importance to the field.

- The paper is well-structured, and its theoretical framework is rigorous.

**Weaknesses:**

- The introduction cites recent non-linear RNN works (e.g., LaCT [1]) but lacks a detailed discussion of the relationship between the proposed method and these existing approaches. Clarifying these connections is crucial for readers to contextualize and understand the paper's novel contributions.
- The authors should explicitly differentiate their proposed state normalization from the Fast-weight normalization used in LaCT [1]. Furthermore, for the orthogonal update mechanism, a comparison (either theoretical or empirical) of the pros and cons of the proposed method against LaCT's Muon would greatly benefit the reader's understanding.
- In the experiments, it is unclear if the chunk_size used is consistent with that in LaCT [1]. LaCT reported significant GPU efficiency issues when chunk_size=1. The authors should clarify if their method shares this limitation. A practical efficiency comparison (e.g., training speed, memory footprint) against baselines like TTT [2], LaCT, and the proposed Lattice method is necessary.

[1] Test-Time Training Done Right.

[2] Learning to (Learn at Test Time):RNNs with Expressive Hidden States

**Questions:**

- There appears to be a discrepancy regarding the LAMBADA dataset. It is mentioned in the text (line 460), but its performance results are missing from Table 2.
- The experiments are primarily focused on commonsense reasoning. To better demonstrate the potential and generalizability of the proposed Lattice as a foundational LLM architecture, could the authors provide results on a wider range of tasks (e.g., vision, time-series, or biomedical data)?

---

> ### Author Response · Authors · 2025-11-24
>
> We thank the reviewer for their valuable feedback and for recognizing the significance of our architectural design.
>
> - **Relationship with LaCT and Muon:** We acknowledge the new method LaCT, which is a concurrent work.  We have added a detailed discussion in the revised paper. Below, we summarize the key distinctions:
>
>
>    - Lattice focuses on a principled memory update mechanism. We derive an interpretable, non-redundant update rule from the online optimization of a state-normalized reconstruction loss.   In contrast, LaCT focuses primarily on hardware efficiency. It proposes maximizing hardware utilization via very large chunk sizes, which facilitates the use of computationally intensive optimizers like Muon. Moreover, it integrates local window attention alongside TTT layers to model locality within these large chunks.
>
>
>    - **Orthogonal Update Mechanism vs. Muon:** Our Orthogonal State Recurrence (OSR) *projects the input onto the orthogonal complement of the current state* ($h_t^{\perp s_{t-1}}$). The objective of OSR  is compression efficiency—ensuring that we only store novel, non-redundant information in its limited space.  In LaCT, however, Muon performs spectral normalization (via Newton-Schulz orthogonalization) on the *gradient matrix*, resulting in $Muon(G) \approx U V^T$ where $U$, $V$ are orthogonal singular vectors. This spectral preconditioner generally improves the convergence speed. While Muon introduces computational overhead, it becomes less significant due to LaCT's very large chunk sizes.
> Consequently, Muon’s spectral normalization of the gradient matrix is fundamentally different from the relative orthogonal projection of the input w.r.t. state, utilized in OSR.
>
>
> - **State Normalization Differences**:
>   - *Lattice (Intra-step Stability)*: We apply normalization *per update step (Eq. 10)*. This is structurally necessary to prevent state explosion caused by the Pythagorean addition inherent in orthogonal updates . As analytically shown in Proposition 3.1,  this provides a geometric grounding, proving that our update is equivalent to optimization on a Riemannian manifold with retraction. The detailed integration of this normalization into the intra-chunk update is provided in Section 3.4 (Eqs. 12–16) and Appendix B.2.1.
>
>   - *LaCT (Inter-chunk Normalization)*: In contrast, LaCT typically applies normalization after each chunk. A key distinction is that LaCT's recurrent update and read-out within a chunk use the un-normalized state, while normalization is applied only at *chunk boundaries*.  This can potentially cause a discrepancy between intra-chunk and inter-chunk dynamics, which Lattice avoids via continuous constraint satisfaction.
> - **Efficiency and Chunk Size**
>   - We confirm that chunk_size=1 is memory-bound and slow on accelerators (TPUs/GPUs). However, as noted in the paper, our main language modeling results utilize a chunk size of $C=4$.
>   - *Parallelization Strategy:* We address efficiency by deriving a specific parallel chunk-wise form for our non-linear recurrence in Section 3.4.1 (Eq. 13 and 15), where we show two approximations for this recurrent method. The chunk-wise form in (16) can be formulated as the linear recurrent update with vectorized forget gate which utilizes a matmul-optimal parallel form of gated linear attention (GLA) (Zhang et al., 2024).  We included the code for Lattice update rule based on these parallelization methods and also point to a throughput analysis for chunk size of 64 in Appendix C.
>   - We view LaCT's hardware-centric techniques (large chunks + Muon) and Lattice's mechanism-centric approach (orthogonal recurrence + state normalization) as complementary. Our primary objective was to establish the Orthogonal State Recurrence as an expressive compression mechanism. As noted in our Limitations section, future work can tailor hardware-efficiency optimization  techniques, such as LaCT's large-chunk for test time training, to Lattice to further optimize its runtime efficiency .
>
>
>
> - **Dataset Citation Discrepancy:** We thank the reviewer for catching this discrepancy. We have corrected the text in the revised manuscript to align with the reported tables.
>
>
> > *“The experiments are primarily focused on commonsense reasoning”*
>
> To demonstrate the model's capability beyond commonsense reasoning benchmarks, we included new results for the Multi-Query Associative Recall (MQAR) task in Appendix C. This is a recall-intensive task designed to test memory capacity. Lattice demonstrates significant improvements in recall accuracy compared to baselines, particularly TTT. These results highlight the efficacy of Lattice's compression mechanism in storing and retrieving dense information over long contexts.

---

> > ### Comment · Reviewer_SnZw · 2025-11-27
> >
> > Thank you for the authors' timely response. However, I have a few remaining inquiries:
> >
> > - Conceptual Alignment with LaCT: While the motivations differ, are the state $S$ in the Lattice framework and the model weights updated by Muon in LaCT fundamentally the same entity within the broader TTT context? If they are consistent, what are the essential differences between the Orthogonal State Recurrence (OSR) and Muon updates, beyond their differing motivations? Clarifying this distinction would greatly aid reader understanding.
> >
> > - Per-token vs. Per-chunk Trade-off: Although Lattice features a carefully designed per-token update and normalization scheme, could the authors provide experimental evidence quantifying the performance benefits of per-token updates compared to a per-chunk approach? Given that the per-token implementation is significantly more complex and the current approximation method limits the scalability of the chunk size (thereby hindering further GPU utilization improvements), it is crucial to justify this complexity with empirical gains.
> >
> > - Generalization: Beyond NLP, could the authors provide experiments on other sequence data modalities, such as computer vision tasks?

---

> ### Author Response · Authors · 2025-12-03
> **Clarification on Essential Differences with LaCT and Per-token vs. Per-chunk Normalization Trade-off**
>
> We thank the reviewer for the prompt engagement and follow-up questions.
>
>
>
>
> 1. **Conceptual Alignment & Essential Differences with LaCT**:
> Within the broader TTT context, the state $S_t$ in Lattice and the fast weights in LaCT are both the compressed key-value memory matrix updated by online optimization. It is worth noting that within this context,  most recurrent models (e.g., Mamba2, DeltaNet, RWKV) can also be formulated as online learners of a reconstruction loss. However, the Orthogonal State Recurrence (OSR) and Muon updates are fundamentally distinct in *what* they orthogonalize and *why*:
>
>
>
>
>     - *OSR* projects the gradient (input token ($h_t$)) onto the orthogonal complement of the current state ($S_{t-1}$):  $\Delta s \propto h_t^{\perp s_{t-1}}$. This is a capacity management mechanism that acts as a filter to ensure only "novel" (non-redundant) information is written to memory.
>
>
>
>
>     - *Muon*, in contrast, applies Newton-Schulz iterations to *the gradient matrix itself* to normalize its spectral spectrum:  $\Delta s = Muon(G) \approx U V^T $. This is an optimization preconditioner technique normally used to speed up convergence.
>
>
>
>
>
>
>
>
> 2. **Normalization: Per-token vs. Per-chunk Trade-off**
>     - *Computational Complexity:* As detailed in Appendix B.2.1 (sec. Computing $\beta$), we showed that the computation of per-token normalization can be simplified significantly, relying only on element-wise operations and summation, without requiring the materialization of $\Delta S_\tau$ (or heavy matrix-matrix multiplications). Consequently, the computational complexity is strictly $\mathcal{O}(CM)$, as evidenced by the code (lines 89-93) in Appendix D.
>     - *Empirical Results:* The table below compares the perplexity of the chunkwise approximation (Eq. 13) using per-token vs. per-chunk normalization strategies:
> | Model | Normalization | Perplexity |
> |:---|:---|:---:|
> | Lattice (C=4) | Per-token | 15.15 |
> | Lattice (C=4) | Per-chunk | 15.17 |
> | Lattice (C=16) | Per-token | 15.32 |
> | Lattice (C=16) | Per-chunk | 15.36 |
>
>
>     - The results show that the divergence grows with larger chunks. Furthermore, for the simplified approximation (Eq. 15) with $C=16$, we observed that per-chunk normalization led to unstable training. These results highlight the importance of per-token normalization for performance and training stability with larger chunk sizes. As discussed, applying normalization only at chunk boundaries can cause a discrepancy between intra-chunk and inter-chunk update dynamics.
>     - In summary, it is structurally necessary to perform per-token normalization to prevent the state explosion caused by the orthogonal addition.  This necessity is specific to our approach and does not apply to TTT update rules in general. For example, in DeltaNet (a TTT with linear layer), the norm of each eigenvalue of the transition matrices does not exceed one; hence, the state remains bounded naturally without this explicit per-token normalization.
>
>
>
>
> 3. **Generalization (Vision/Time-series)**: Our current focus is on Language Modeling, where the "quadratic bottleneck" of attention and the need for long-context recall are most critical. This also aligns with the evaluation protocols used in baseline works in efficient sequence modeling (such as DeltaNet , Gated DeltaNet , and TTT), which similarly focus on language benchmarks. Furthermore, to demonstrate generalization capabilities beyond standard perplexity metrics, we included the Recall-Intensive MQAR task in Appendix C to evaluate the model's fundamental capacity to store and retrieve dense sequence information. While we believe the OSR mechanism would naturally extend to Time-Series and Vision, we respectfully suggest that our extensive benchmarking constitutes a comprehensive validation of the capabilities of the proposed architecture.  Extending this work to other modalities is beyond the limitation of the rebuttal period and would be the topic of future research, similar to how the extension of TTT/GLA to the video/vision domain has been studied in separate works like [4] and [5].
>
>
> References:
> [4] Dalal, Karan, et al. "One-minute video generation with test-time training." Proceedings of the Computer Vision and Pattern Recognition Conference. 2025.
> [5] Liao, Bencheng, et al. "Vig: Linear-complexity visual sequence learning with gated linear attention." Proceedings of the AAAI Conference on Artificial Intelligence. Vol. 39. No. 5. 2025.

---

### Official Review · Reviewer_9GZ4 · 2025-11-04

**Soundness:** 3
**Presentation:** 3
**Contribution:** 3
**Rating:** 8
**Confidence:** 2

**Summary:**

The paper introduces Lattice, a new architecture that replaces transformer attention with a compressed memory representation. Instead of a growing K-V cache they use a streaming RNN with a fixed number of 'memory slots'. Each new token updates a small set of memory slots by writing only the orthogonal part of its information, enabling efficient streaming with constant memory.

Across many language-modeling benchmark tasks, Lattice matches or achieves lower perplexity than state-of-the-art transformers and modern linear RNNs.

**Strengths:**

By removing the growing KV cache they successfully avoid a problem with standard transformers at long contexts.

Results are competitive with state-of-the-art transformers and SSMs.

**Weaknesses:**

Hard to know if the model will continue to outperform transformers at much larger parameter scales and very long contexts.

Given that the fixed number of memory slots is so important to this new architecture, it is surprising that the paper does not discuss how the size of this dimension is selected or varied in experiments.

It is not clear from the paper how much Lattice affects training time or whether it increases compute requirements at inference.

minor typo:
1174 "final learning rate of 3e5" -> 3e-5

**Questions:**

I wasn't entirely sure how you choose the number of memory slots, how do you pick this hyperparameter? How sensitive is performance to your choice of the number of memory slots?

A table with the choice of number of memory slots and performance would be interesting to see.

Could you provide information about wall-clock training time in your comparisons?

---

> ### Author Response · Authors · 2025-11-24
>
> We thank the reviewer for their constructive feedback and positive assessment, and for recognizing the importance and novelty of the proposed approach. Below, we address your specific suggestions/comments:
>
>
> - **Selection and Sensitivity of Memory Slots ($m$):** For our main experiments, we set $m=d$ (head dimension) for all models. To address the sensitivity to memory size, we conducted a new ablation study (added to the revised paper in Table 3 and Appendix C). We observe that performance scales linearly with $\log(m)$, indicating a consistent and predictable gain in expressivity as memory capacity increases. Remarkably, \model with only 16 memory slots ($m=d/4=16$) outperforms TTT with full memory ($m=64$). This result is significant: it empirically validates that the orthogonal update mechanism utilizes state capacity significantly more efficiently than standard updates , allowing for high performance even with smaller $m$ values.
>
> - **Training Time and Model Throughput:** We have provided a wall-clock throughput comparison in Figure 6 demonstrating that Lattice share a similar pattern as recurrent models in general for throughput scalability and exhibit superior throughput scalability compared to Transformers as sequence length increases. Regarding the performance retention, as Figure 2 (Left) shows, the performance gap between Lattice and baselines widens as sequence length increases (e.g., from 2k to 16k).   Consistent with other recurrent models, Lattice models take advantage of constant inference compute per token ($\mathcal{O}(1)$) and fixed memory usage ($\mathcal{O}(m)$), regardless of sequence length.
>
> -**Typo Correction:** We thank the reviewer for catching the typo, which has been corrected in the revised manuscript.

---

### Comment · Area_Chair_cWyZ · 2025-11-26
**A Reminder on Your Crucial Role in the ICLR Discussion Period**

Dear Reviewers:

As the Area Chair, I would like to sincerely thank you for the time and expertise you have invested in writing your initial review. Your insights are invaluable to the decision-making process.

We are now entering the critical discussion and rebuttal phase. This is a collaborative process where authors have the opportunity to address your concerns and questions. Your active participation in this phase is essential to ensure we reach a fair and well-informed final decision.

I strongly encourage you to:

Engage with the Authors' Rebuttal: Please read the authors' response carefully and substantively.

Participate in the Discussion: Engage with the other reviewers on the forum. If the authors have clarified a point, please acknowledge it. If you have follow-up questions or remaining concerns, please voice them. Your dialogue with fellow reviewers is key to reaching a consensus.

Update Your Review (if necessary): Based on the discussion and rebuttal, you may feel the need to adjust your score or final recommendation. Please do so, as it reflects a more holistic view of the paper.

Your continued engagement ensures the integrity and quality of the ICLR conference. Thank you for your vital contribution to our community.

Best regards,

Area Chair, ICLR 2026

---

### Author Response · Authors · 2025-12-03
**Summary of Revisions and Additional Experiments**

We thank the reviewers for their constructive feedback. In response, we have updated the manuscript with significant new experimental results, theoretical clarifications, and implementation details. Below is a summary of the key changes:

1. **New Experimental Results**
   - **Memory Capacity Ablation (Figure 6):** We conducted a new ablation study scaling the memory size ($m$). Results demonstrate that Lattice with only 16 slots ($m=d/4$) outperforms the TTT baseline with 64 slots ($m=d$). This empirically validates that the orthogonal update mechanism utilizes state capacity significantly more efficiently than standard updates *[Reviewers 9GZ4, ViSx, PU48]*.
   - **Recall-Intensive Task (MQAR)  (Figure 7):** We added the Multi-Query Associative Recall (MQAR) benchmark to evaluate memory capacity under stress. Lattice demonstrates significant improvements in recall accuracy compared to TTT and DeltaNet, particularly as sequence length increases, highlighting its robustness against memory interference *[Reviewer PU48, Cao9, SnZw]*.
   - **Throughput Comparison ( Figure 8):** Throughput for 110M parameter models trained using a chunk Size of 64 and 16 *[Reviewers Cao9, 9GZ4, ViSx]*.
   - **Per-token vs. Per-chunk Normalization Ablation (Table 5):** We compared the perplexity of the chunk-wise approximation using per-token versus per-chunk normalization strategies. These results highlight the importance of per-token normalization for ensuring both performance and stability with larger chunk sizes *[Reviewer SnZw]*.

2. **Theoretical Clarifications & Comparisons**
   - **Relationship with LaCT & Muon:** We added a detailed discussion distinguishing Lattice from the concurrent work LaCT (Test-Time Training Done Right) in Discussion Section *[Reviewer SnZw]*.
   - **Comparison with RWKV:** We expanded Table 1 and the results section to include comparisons with the RWKV family (RWKV-6, RWKV-7), ensuring a more comprehensive evaluation against state-of-the-art linear RNNs.  Also we included performance comparison to RWKV-5/6 in Figure 7 *[Reviewer Cao9]*.
   - **Throughput Comparison:** We clarified the throughput comparison on TPUs, where Lattice exhibits linear scalability with respect to sequence length, significantly outperforming Transformers on long contexts (e.g., 16k tokens) *[Reviewer Cao9]*.

3. **Implementation:** We included a JAX/Flax code of the Lattice update rule (both step-wise and parallel chunk-wise) *[Reviewer ViSx, PU48]*.

4. **Manuscript Improvements:** We updated Limitation and Future Work paragraph and
corrected the typos. *[Reviewers Cao, SnZw,9GZ4]*.

---

### Meta-Review · Area_Chair_p2uq · 2026-01-08

**Summary:**

The method formulates sequence modeling as an online optimization problem, deriving a dynamic update rule that projects inputs onto the orthogonal complement of the current state. The review process was active, with the authors providing a substantial rebuttal. However, significant concerns remain regarding the significance of the contribution and practical efficiency.

**Reviewer Concerns:**

A primary concern is that Lattice appears to be an incremental modification of the TTT framework.

The proposed technique introduces a dependency that complicates parallelization.

The added complexity of the orthogonal update rule gets marginal gains.

**Reviewer Scores:**

The review process was active, with the authors providing a substantial rebuttal. However, significant concerns remain regarding the significance of the contribution and practical efficiency.

---

### Decision · Program_Chairs · 2026-01-26

Reject